# Best of Both Worlds: Practical and Theoretically Optimal Submodular Maximization in Parallel

**Yixin Chen**
Department of Computer Science
Florida State University
Tallahassee, Florida
yc19e@my.fsu.edu

**Tonmoy Dey**
Department of Computer Science
Florida State University
Tallahassee, Florida
td18d@my.fsu.edu

**Alan Kuhnle**
Department of Computer Science
Florida State University
Tallahassee, Florida
kuhnle@cs.fsu.edu

## Abstract

For the problem of maximizing a monotone, submodular function with respect to a cardinality constraint $k$ on a ground set of size $n$, we provide an algorithm that achieves the state-of-the-art in both its empirical performance and its theoretical properties, in terms of adaptive complexity, query complexity, and approximation ratio; that is, it obtains, with high probability, query complexity of $O(n)$ in expectation, adaptivity of $O(\log(n))$, and approximation ratio of nearly $1 - 1/e$. The main algorithm is assembled from two components which may be of independent interest. The first component of our algorithm, LINEARSEQ, is useful as a preprocessing algorithm to improve the query complexity of many algorithms. Moreover, a variant of LINEARSEQ is shown to have adaptive complexity of $O(\log(n/k))$ which is smaller than that of any previous algorithm in the literature. The second component is a parallelizable thresholding procedure THRESHOLDSEQ for adding elements with gain above a constant threshold. Finally, we demonstrate that our main algorithm empirically outperforms, in terms of runtime, adaptive rounds, total queries, and objective values, the previous state-of-the-art algorithm FAST in a comprehensive evaluation with six submodular objective functions.

## 1 Introduction

The cardinality-constrained optimization of a monotone, submodular function $f : 2^{\mathcal{N}} \to \mathbb{R}^+$, defined on subsets of a ground set $\mathcal{N}$ of size $n$, is a general problem formulation that is ubiquitous in wide-ranging applications, *e.g.* video or image summarization [30], network monitoring [26], information gathering [23], and MAP Inference for Determinantal Point Processes [20], among many others. The function $f : 2^{\mathcal{N}} \to \mathbb{R}^+$ is submodular iff for all $S \subseteq T \subseteq \mathcal{N}$, $x \notin T$, $\Delta(x \mid T) \leq \Delta(x \mid S)$[1]; and the function $f$ is monotone if $f(S) \leq f(T)$ for all $S \subseteq T$. In this paper, we study the following submodular maximization problem (SM)

$$\text{maximize} f(S), \text{ subject to } |S| \leq k, \tag{SM}$$

where $f$ is a monotone, submodular function; SM is an NP-hard problem. There has been extensive effort into the design of approximation algorithms for SM over the course of more than 45 years, *e.g.* [32, 12, 10, 21, 24]. For SM, the optimal ratio has been shown to be $1 - 1/e \approx 0.63$ [32].

---

[1] $\Delta(x \mid S)$ denotes the *marginal gain* of $x$ to $S$: $f(S \cup \{x\}) - f(S)$.

35th Conference on Neural Information Processing Systems (NeurIPS 2021).

Table 1: Theoretical comparison to algorithms that achieve nearly optimal adaptivity, ratio, and query complexity: the algorithm of Ene and Nguyen [14], RANDOMIZED-PARALLEL-GREEDY (RPG) of Chekuri and Quanrud [11], BINARY-SEARCH-MAXIMIZATION (BSM) and SUBSAMPLE-MAXIMIZATION (SM) of Fahrbach et al. [17], FAST of Breuer et al. [9]. The symbol † indicates the result holds with constant probability or in expectation; the symbol ‡ indicates the result does not hold on all instances of SM; while no symbol indicates the result holds with probability greater than $1 - O(1/n)$. Observe that our algorithm LS+PGB dominates in at least one category when compared head-to-head with any one of the previous algorithms.

| Reference | Ratio | Adaptivity | Queries |
|---|---|---|---|
| Ene and Nguyen [14] | $1 - 1/e - \varepsilon$ | $O\left(\frac{1}{\varepsilon^2}\log(n)\right)$ | $O\left(n\mathrm{poly}(\log n, 1/\varepsilon)\right)$ |
| Chekuri and Quanrud [11] (RPG) | $1 - 1/e - \varepsilon$ | $O\left(\frac{1}{\varepsilon^2}\log(n)\right)$ † | $O\left(\frac{n}{\varepsilon^4}\log(n)\right)$ † |
| Fahrbach et al. [17] (BSM) | $1 - 1/e - \varepsilon$ † | $O\left(\frac{1}{\varepsilon^2}\log(n)\right)$ | $O\left(\frac{n}{\varepsilon^3}\log\log k\right)$ † |
| Fahrbach et al. [17] (SM) | $1 - 1/e - \varepsilon$ † | $O\left(\frac{1}{\varepsilon^2}\log(n)\right)$ | $O\left(\frac{n}{\varepsilon^3}\log(1/\varepsilon)\right)$ † |
| Breuer et al. [9] (FAST) | $1 - 1/e - \varepsilon$ ‡ | $O\left(\frac{1}{\varepsilon^2}\log(n)\log^2\left(\frac{\log(k)}{\varepsilon}\right)\right)$ | $O\left(\frac{n}{\varepsilon^2}\log\left(\frac{\log(k)}{\varepsilon}\right)\right)$ |
| LS+PGB [Theorem 3] | $1 - 1/e - \varepsilon$ | $O\left(\frac{1}{\varepsilon^2}\log(n/\varepsilon)\right)$ | $O\left(\frac{n}{\varepsilon^2}\right)$ † |

As instance sizes have grown very large, there has been much effort into the design of efficient, parallelizable algorithms for SM. Since queries to the objective function can be very expensive, the overall efficiency of an algorithm for SM is typically measured by the *query complexity*, or number of calls made to the objective function $f$ [2, 9]. The degree of parallelizability can be measured by the *adaptive complexity* of an algorithm, which is the minimum number of rounds into which the queries to $f$ may be organized, such that within each round, the queries are independent and hence may be arbitrariliy parallelized. Observe that the lower the adaptive complexity, the more parallelizable an algorithm is. To obtain a constant approximation factor, a lower bound of $\Omega(n)$ has been shown on the query complexity [24] and a lower bound of $\Omega(\log(n)/\log\log(n))$ has been shown on the adaptive complexity [3].

Several algorithms have been developed recently that are nearly optimal in terms of query and adaptive complexities [14, 11, 17, 5]; that is, these algorithms achieve $O(\log n)$ adaptivity and $O(n\mathrm{polylog}(n))$ query complexity (see Table 1). However, these algorithms use sampling techniques that result in very large constant factors that make these algorithms impractical. This fact is discussed in detail in Breuer et al. [9]; as an illustration, to obtain ratio $1 - 1/e - 0.1$ with 95% confidence, all of these algorithms require more than $10^6$ queries of sets of size $k/\log(n)$ in every adaptive round [9]; moreover, even if these algorithms are run as heuristics using a single sample, other inefficiencies preclude these algorithms of running even on moderately sized instances [9]. For this reason, the FAST algorithm of Breuer et al. [9] has been recently proposed, which uses an entirely different sampling technique called adaptive sequencing. Adaptive sequencing was originally introduced in Balkanski et al. [6], but the original version has quadratic query complexity in the size of the ground set and hence is still impractical on large instances. To speed it up, the FAST algorithm sacrifices theoretical guarantees to yield an algorithm that parallelizes well and is faster than all previous algorithms for SM in an extensive experimental evaluation. The theoretical sacrifices of FAST include: the adaptivity of FAST is $\Omega(\log(n)\log^2(\log n))$, which is higher than the state-of-the-art, and more significantly, the algorithm obtains no approximation ratio for $k < 850^2$; since many applications require small choices for $k$, this limits the practical utility of FAST. A natural question is thus: *is it possible to design an algorithm that is both practical and theoretically optimal in terms of adaptivity, ratio, and total queries?*

## 1.1 Contributions

In this paper, we provide three main contributions. The first contribution is the algorithm LINEARSEQ (LS, Section 2) that achieves with probability $1 - 1/n$ a constant factor $(4 + O(\varepsilon))^{-1}$ in expected linear query complexity and with $O(\log n)$ adaptive rounds (Theorem 1). Although the ratio of $\approx 0.25$ is smaller than the optimal $1 - 1/e \approx 0.63$, this algorithm can be used to improve the query

---

[2]The approximation ratio $1 - 1/e - 4\varepsilon$ of FAST holds with probability $1 - \delta$ for $k \geq \theta(\varepsilon, \delta, k) = 2\log(2\delta^{-1}\log(\frac{1}{\varepsilon}\log(k)))/\varepsilon^2(1 - 5\varepsilon)$.

Table 2: Empirical comparison of our algorithm with FAST, with each algorithm using 75 CPU threads on a broad range of $k$ values and six applications; details provided in Section 4. Parameter settings are favorable to FAST, which is run without theoretical guarantees while LS+PGB enforces ratio $\approx 0.53$ with high probability. For each application, we report the arithmetic mean of each metric over all instances. Observe that LS+PGB outperforms FAST in runtime on all applications.

| | Runtime (s) | | Objective Value | | Queries | |
|---|---|---|---|---|---|---|
| Application | FAST | LS+PGB | FAST | LS+PGB | FAST | LS+PGB |
| TrafficMonitor | $3.7 \times 10^{-1}$ | $\mathbf{2.1 \times 10^{-1}}$ | $4.7 \times 10^8$ | $\mathbf{5.0 \times 10^8}$ | $3.5 \times 10^3$ | $\mathbf{2.4 \times 10^3}$ |
| InfluenceMax | $4.4 \times 10^0$ | $\mathbf{2.3 \times 10^0}$ | $1.1 \times 10^3$ | $1.1 \times 10^3$ | $7.7 \times 10^4$ | $\mathbf{4.0 \times 10^4}$ |
| TwitterSumm | $1.6 \times 10^1$ | $\mathbf{3.5 \times 10^0}$ | $3.8 \times 10^5$ | $3.8 \times 10^5$ | $1.5 \times 10^5$ | $\mathbf{6.2 \times 10^4}$ |
| RevenueMax | $3.9 \times 10^2$ | $\mathbf{5.4 \times 10^1}$ | $1.4 \times 10^4$ | $1.4 \times 10^4$ | $7.6 \times 10^4$ | $\mathbf{2.7 \times 10^4}$ |
| MaxCover (BA) | $3.7 \times 10^2$ | $\mathbf{7.6 \times 10^1}$ | $6.0 \times 10^4$ | $\mathbf{6.3 \times 10^4}$ | $5.8 \times 10^5$ | $\mathbf{1.8 \times 10^5}$ |
| ImageSumm | $1.6 \times 10^1$ | $\mathbf{8.1 \times 10^{-1}}$ | $9.1 \times 10^3$ | $9.1 \times 10^3$ | $1.3 \times 10^5$ | $\mathbf{4.8 \times 10^4}$ |

complexity of many extant algorithms, as we decribe in the related work section below. Interestingly, LINEARSEQ can be modified to have adaptivity $O\left(\log(n/k)\right)$ at a small cost to its ratio as discussed in Appendix F. This version of LINEARSEQ is a constant-factor algorithm for SM with smaller adaptivity than any previous algorithm in the literature, especially for values of $k$ that are large relative to $n$.

Our second contribution is an improved parallelizable thresholding procedure THRESHOLDSEQ (TS, Section 3) for a commonly recurring task in submodular optimization: namely, add all elements that have a gain of a specified threshold $\tau$ to the solution. This subproblem arises not only in SM, but also *e.g.* in submodular cover [17] and non-monotone submodular maximization [4, 18, 15, 25]. Our TS accomplishes this task with probability $1 - 1/n$ in $O(\log n)$ adaptive rounds and expected $O(n)$ query complexity (Theorem 2), while previous procedures for this task only add elements with an expected gain of $\tau$ and use expensive sampling techniques [17]; have $\Omega(\log^2 n)$ adaptivity [22]; or have $\Omega(kn)$ query complexity [6].

Finally, we present in Section 3 the parallelized greedy algorithm PARALLELGREEDYBOOST (PGB), which is used in conjunction with LINEARSEQ and THRESHOLDSEQ to yield the final algorithm LS+PGB, which answers the above question affirmatively: LS+PGB obtains nearly the optimal $1 - 1/e$ ratio with probability $1 - 2/n$ in $O(\log n)$ adaptive rounds and $O(n)$ queries in expectation; moreover, LS+PGB is faster than FAST in an extensive empirical evaluation (see Table 2). In addition, LS+PGB improves theoretically on the previous algorithms in query complexity while obtaining nearly optimal adaptivity (see Table 1).

## 1.2 Additional Related Work

**Adaptive Sequencing.** The main inefficiency of the adaptive sequencing method of Balkanski et al. [6] (which causes the quadratic query complexity) is an explicit check that a constant fraction of elements will be filtered from the ground set. In this work, we adopt a similar sampling technique to adaptive sequencing, except that we design the algorithm to filter a constant fraction of elements with only constant probability. This method allows us to reduce the quadratic query complexity of adaptive sequencing to linear query complexity while only increasing the adaptive complexity by a small constant factor. In contrast, FAST of Breuer et al. [9] speeds up adaptive sequencing by increasing the adaptive complexity of the algorithm through adaptive binary search procedures, which, in addition to the increasing the adaptivity by logarithmic factors, place restrictions on the $k$ values for which the ratio can hold. This improved adaptive sequencing technique is the core of our THRESHOLDSEQ procedure, which has the additional benefit of being relatively simple to analyze.

**Algorithms with Linear Query Complexity.** Our LINEARSEQ algorithm also uses the improved adaptive sequencing technique, but in addition this algorithm integrates ideas from the $\Omega(n)$-adaptive linear-time streaming algorithm of Kuhnle [24] to achieve a constant-factor algorithm with low adaptivity in expected linear time. Integration of the improved adaptive sequencing with the ideas of Kuhnle [24] is non-trivial, and ultimately this integration enables the theoretical improvement in query complexity over previous algorithms with sublinear adaptivity that obtain a constant ratio with

high probability (see Table 1). In Fahrbach et al. [17], a linear-time procedure SUBSAMPLEPRE-PROCESSING is described; this procedure is to the best of our knowledge the only algorithm in the literature that obtains a constant ratio with sublinear adaptive rounds and linear query complexity and hence is comparable to LINEARSEQ. However, SUBSAMPLEPREPROCESSING uses entirely different ideas from our LINEARSEQ and has much weaker theoretical guarantees: for input $0 < \delta < 1$, it obtains ratio $\frac{\delta^2}{2 \times 10^6}$ with probability $1 - \delta$ in $O(\log(n)/\delta)$ adaptive rounds and $O(n)$ queries in expectation – the small ratio renders SUBSAMPLEPREPROCESSING impractical; also, its ratio holds only with constant probability. By contrast, with $\varepsilon = 0.1$, our LINEARSEQ obtains ratio $\approx 0.196$ with probability $1 - 1/n$ in $O(\log(n))$ adaptive rounds and $O(n)$ queries in expectation.

**Using LS for Preprocessing: Guesses of OPT.** Many algorithms for SM, including FAST and all of the algorithms listed in Table 1 except for SM and our algorithm, use a strategy of guessing logarithmically many values of OPT. Our LINEARSEQ algorithm reduces the interval containing OPT from size $k$ to a small constant size in expected linear time. Thus, LINEARSEQ could be used for preprocessing prior to running FAST or one of the other algorithms in Table 1, which would improve their query complexity without compromising their adaptive complexity or ratio; this illustrates the general utility of LINEARSEQ. For example, with this change, the theoretical adaptivity of FAST improves, although it remains worse than LS+PGB: the adaptive complexity of FAST becomes $O\left(\frac{1}{\varepsilon^2} \log(n) \log\left(\frac{1}{\varepsilon} \log(k)\right)\right)$ in contrast to the $O\left(\frac{1}{\varepsilon^2} \log(n/\varepsilon)\right)$ of LS+PGB. Although SUBSAMPLEPREPROCESSING may be used for the same purpose, its ratio only holds with constant probability which would then limit the probability of success of any following algorithm.

**Relationship of THRESHOLDSEQ to Existing Methods.** The first procedure in the literature to perform the same task is the THRESHOLDSAMPLING procedure of Fahrbach et al. [17]; however, THRESHOLDSAMPLING only ensures that the *expected* marginal gain of each element added is at least $\tau$ and has large constants in its runtime that make it impractical [9]. In contrast, THRESHOLDSEQ ensures that added elements contribute a gain of at least $\tau$ *with high probability* and is highly efficient empirically. A second procedure in the literature to perform the same task is the ADAPTIVE-SEQUENCING method of Balkanski et al. [6], which similarly to THRESHOLDSEQ uses random permutations of the ground set; however, ADAPTIVE-SEQUENCING focuses on explicitly ensuring a constant fraction of elements will be filtered in the next round, which is expensive to check: the query complexity of ADAPTIVE-SEQUENCING is $O(kn)$. In contrast, our THRESHOLDSEQ algorithm ensures this property with a constant probability, which is sufficient to ensure the adaptivity with the high probability of $1 - 1/n$ in $O(n)$ expected queries. Finally, a third related procedure in the literature is THRESHOLDSAMPLING of Kazemi et al. [22], which also uses random permutations to sample elements. However, this algorithm has the higher adaptivity of $O(\log(n) \log(k))$, in contrast to the $O(\log(n))$ of THRESHOLDSEQ.

**MapReduce Framework.** Another line of work studying parallelizable algorithms for SM has focused on the MapReduce framework [13] in a distributed setting, *e.g.* [7, 8, 16, 28]. These algorithms divide the dataset over a large number of machines and are intended for a setting in which the data does not fit on a single machine. None of these algorithms has sublinear adaptivity and hence all have potentially large numbers of sequential function queries on each machine. In this work, our empirical evaluation is on a single machine with a large number of CPU cores; we do not evaluate our algorithms in a distributed setting.

**Organization.** The constant-factor algorithm LINEARSEQ is described and analyzed in Section 2; the details of the analysis are presented in Appendix C. The variant of LINEARSEQ with lower adaptivity is described in Appendix F. The algorithms THRESHOLDSEQ and PARALLELGREEDYBOOST are discussed at a high level in Section 3, with detailed descriptions of these algorithms and theoretical analysis presented in Appendices D and E. Our empirical evaluation is summarized in Section 4 with more results and discussion in Appendix H.

## 2  A Parallelizable Algorithm with Linear Query Complexity: LINEARSEQ

In this section, we describe the algorithm LINEARSEQ for SM (Alg. 1) that obtains ratio $(4 + O(\varepsilon))^{-1}$ in $O\left(\frac{1}{\varepsilon^3} \log(n)\right)$ adaptive rounds and expected $O\left(\frac{n}{\varepsilon^3}\right)$ queries. If $\varepsilon \leq 0.21$, the ratio of LINEARSEQ is lower-bounded by $(4 + 16\varepsilon)^{-1} \geq 0.135$, which shows that a relatively large constant ratio is obtained even at large values of $\varepsilon$. An initial run of this algorithm is required for our main algorithm LS+PGB.

**Algorithm 1** The algorithm that obtains ratio $(4 + O(\varepsilon))^{-1}$ in $O\left(\log(n)/\varepsilon^3\right)$ adaptive rounds and expected $O\left(n/\varepsilon^3\right)$ queries.

---

1: **procedure** LINEARSEQ($f, \mathcal{N}, k, \varepsilon$)
2:     **Input:** evaluation oracle $f : 2^{\mathcal{N}} \to \mathbb{R}^+$, constraint $k$, error $\varepsilon$
3:     $a = \arg\max_{u \in \mathcal{N}} f(\{u\})$
4:     Initialize $A \leftarrow \{a\}$, $V \leftarrow \mathcal{N}$, $\ell = \lceil 4(1 + 1/(\beta\varepsilon))\log(n)\rceil$, $\beta = \varepsilon/(16\log(8/(1 - e^{-\varepsilon/2})))$
5:     **for** $j \leftarrow 1$ to $\ell$ **do**
6:         Update $V \leftarrow \{x \in V : \Delta(x \mid A) \geq f(A)/k\}$ and filter out the rest
7:         **if** $|V| = 0$ **then break**
8:         $V = \{v_1, v_2, \ldots, v_{|V|}\} \leftarrow$ **random-permutation**($V$)
9:         $\Lambda \leftarrow \{\lfloor(1 + \varepsilon)^u\rfloor : 1 \leq \lfloor(1 + \varepsilon)^u\rfloor \leq k, u \in \mathbb{N}\}$
            $\cup\{\lfloor k + u\varepsilon k\rfloor : \lfloor k + u\varepsilon k\rfloor \leq |V|, u \in \mathbb{N}\} \cup \{|V|\}$
10:        $B[\lambda_i] =$ **false**, for $\lambda_i \in \Lambda$
11:       **for** $\lambda_i \in \Lambda$ in parallel **do**
12:          $T_{\lambda_{i-1}} \leftarrow \{v_1, v_2, \ldots, v_{\lambda_{i-1}}\}$ ; $T_{\lambda_i} \leftarrow \{v_1, v_2, \ldots, v_{\lambda_i}\}$ ; $T'_{\lambda_i} \leftarrow T_{\lambda_i} \backslash T_{\lambda_{i-1}}$
13:          **if** $\Delta\left(T'_{\lambda_i} \mid A \cup T_{\lambda_{i-1}}\right)/|T'_{\lambda_i}| \geq (1 - \varepsilon)f(A \cup T_{\lambda_{i-1}})/k$ **then** $B[\lambda_i] \leftarrow$ **true**
14:       $\lambda^* \leftarrow \max\{\lambda_i \in \Lambda : B[\lambda_i] =$ **false** and $((\lambda_i \leq k$ and $B[1]$ to $B[\lambda_{i-1}]$ are all **true**) or
    $(\lambda_i > k$ and $\exists m \geq 1$ s.t. $|\bigcup_{u=m}^{i-1} T'_{\lambda_u}| \geq k$ and $B[\lambda_m]$ to $B[\lambda_{i-1}]$ are all **true**))\}$
15:       $A \leftarrow A \cup T_{\lambda^*}$
16:     **if** $|V| > 0$ **then return** *failure*
17:     **return** $A' \leftarrow$ last $k$ elements added to $A$

---

**Description of LS.** The work of LS is done within iterations of a sequential outer **for** loop (Line 5); this loop iterates at most $O(\log(n))$ times, and each iteration requires two adaptive rounds; thus, the adaptive complexity of the algorithm is $O(\log(n))$. Each iteration adds more elements to the set $A$, which is initially empty. Within each iteration, there are four high-level steps: 1) filter elements from $V$ that have gain less than $f(A)/k$ (Line 6); 2) randomly permute $V$ (Line 8); 3) compute in parallel the marginal gain of adding blocks of the sequence of remaining elements in $V$ to $A$ (**for** loop on Line 11); 4) select a prefix of the sequence $V = (v_1, v_2, \ldots, v_{|V|})$ to add to $A$ (Line 14). The selection of the prefix to add is carefully chosen to approximately satisfy, on average, Condition 1 for elements added; and also to ensure that, with constant probability, a constant fraction of elements of $V$ are filtered on the next iteration.

The following theorem states the theoretical results for LINEARSEQ. The remainder of this section proves this theorem, with intuition and discussion of the proof. The omitted proofs for all lemmata are provided in Appendix C.

**Theorem 1.** *Let $(f, k)$ be an instance of* SM*. For any constant $0 < \varepsilon < 1/2$, the algorithm* LINEARSEQ *has adaptive complexity $O\left(\log(n)/\varepsilon^3\right)$ and outputs $A' \subseteq \mathcal{N}$ with $|A'| \leq k$ such that the following properties hold: 1) The algorithm succeeds with probability at least $1 - 1/n$. 2) There are $O\left((1/(\varepsilon k) + 1)n/\varepsilon^3\right)$ oracle queries in expectation. 3) If the algorithm succeeds, $\left[4 + \frac{4(2 - \varepsilon)}{(1 - \varepsilon)(1 - 2\varepsilon)}\varepsilon\right]f(A') \geq f(O)$, where $O$ is an optimal solution to the instance $(f, k)$.*

**Overview.** The goal of this section is to produce a constant factor, parallelizable algorithm with linear query complexity. As a starting point, consider an algorithm[1] that takes one pass through the ground set, adding each element $e$ to candidate set $A$ iff

$$\Delta(e \mid A) \geq f(A)/k. \tag{1}$$

Condition 1 ensures two properties: 1) the last $k$ elements in $A$ contain a constant fraction of the value $f(A)$; and 2) $f(A)$ is within a constant fraction of OPT. By these two properties, the last $k$ elements of $A$ are a constant factor approximation to SM with exactly one query of the objective function per element of the ground set. For completeness, we give a pseudocode (Alg. 3) and proof in Appendix B. However, each query depends on all of the previous ones and thus there are $n$ adaptive rounds. Therefore, the challenge is to approximately simulate Alg. 3 in a lowly adaptive (highly parallelizable) manner, which is what LINEARSEQ accomplishes.

---

[1]This algorithm is a simplified version of the streaming algorithm of Kuhnle [24].

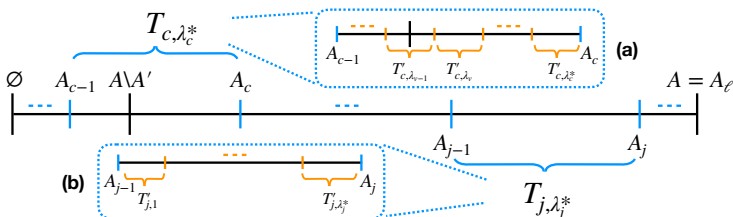

Figure 1: This figure depicts the way elements are added to $A$, with $A_i$ denoting the value of $A$ after iteration $i$ of the outer **for** loop; the prefix added at iteration $i$ is denoted by $T_{i,\lambda_i^*}$. The orange lines mark the blocks comprising each prefix. Also, the set $A'$ of the last $k$ elements added to $A$ is shown. Parts **(a)** and **(b)** depict two separate cases in the proof of Claim 1.

## 2.1 Approximately Satisfying Condition 1

**Discarding Elements.** In one adaptive round during each iteration $j$ of the outer **for** loop, all elements with gain to $A$ of less than $f(A)/k$ are discarded from $V$ (Line 6). Since the size of $A$ increases as the algorithm runs, by submodularity, the gain of these elements can only decrease and hence these elements cannot satisfy Condition 1 and can be safely discarded from consideration. The process of filtering thereby ensures the following lemma at termination.

**Lemma 1.** *At successful termination of* LINEARSEQ, $f(O) \leq 2f(A)$, *where* $O \subseteq \mathcal{N}$ *is an optimal solution of size* $k$.

**Addition of Elements.** Next, we describe the details of how elements are added to the set $A$. The random permutation of remaining elements on Line 8 constructs a sequence $(v_1, v_2, \ldots, v_{|V|})$ such that each element is uniformly randomly sampled from the remaining elements. By testing the marginal gains along the sequence in parallel, it is possible to determine a good prefix of the sequence $(v_i)$ to add to $A$ to ensure the following: 1) Condition 1 is approximately satisfied; and 2) We will discard a constant fraction of $V$ in the next iteration with constant probability. Condition 1 is important for the approximation ratio and discarding a constant fraction of $V$ is important for the adaptivity and query complexity. Below, we discuss how to choose the prefix such that both are achieved. To speed up the algorithm, we do not test the marginal gain at each point in the sequence $(v_i)$, but rather test blocks of elements at once as determined by the index set $\Lambda$ defined in the pseudocode.

**Prefix Selection.** Say a block is *bad* if this block does not satisfy the condition checked on Line 13 (which is an approximate, average form of Condition 1); otherwise, the block is *good*. At the end of an iteration, we select the largest block index $\lambda^*$, where this block is bad and the previous consecutive blocks which together have at least $k$ elements are all good; or this block is bad and all the previous blocks are good blocks. Then, we add the prefix $T_{\lambda^*} = (v_1, v_2, \ldots, v_{\lambda^*})$ into $A$. Now, the relevance of Condition 1 for the approximation ratio is that it implies $f(A) \geq 2f(A \setminus A')$, where $A'$ are the last $k$ elements added to $A$. Lemma 2 shows that the conditions required on the marginal gains of blocks added imply an approximate form of this fact is satisfied by LINEARSEQ. Indeed, the proof of Lemma 2 informs the choice $\Lambda$ of blocks evaluated and the computation of $\lambda^*$.

**Lemma 2.** *Suppose* LINEARSEQ *terminates successfully. Then* $f(A) \geq \frac{2(1-\varepsilon+\varepsilon^2)}{1+\varepsilon} f(A \setminus A')$.

*Proof.* If $|A| \leq k$, the lemma is immediate, so assume $|A| > k$. For iteration $j$, let $T_{j,\lambda_j^*}$ denote the set added to $A$ during iteration $j$; and let $T_{j,\lambda_j^*} = \emptyset$ if the algorithm terminates before iteration $j$. Let $A_j$ denote the value of $A$ after iteration $j$. Define $c = \max\{c \in \mathbb{N} : A' \subseteq (\cup_{j=c}^{\ell} T_{j,\lambda_j^*})\}$. Then, $|T_{c,\lambda_c^*}| > 0$; and for any $j > c$, $|T_{j,\lambda_j^*}| < k$. It holds that $(\cup_{j=c+1}^{\ell} T_{j,\lambda_j^*}) \subset A' \subseteq (\cup_{j=c}^{\ell} T_{j,\lambda_j^*})$. Figure 1 shows how $A$ is composed of these sets $T_{j,\lambda_j^*}$ and how each set is composed of blocks. The following claim is proven in Appendix C.3.

**Claim 1.** *It holds that* $\Delta\left(T_{c,\lambda_c^*} \mid A \setminus A'\right) \geq (1-\varepsilon)\max\{0, |T_{c,\lambda_c^*} \cap A'| - 2\varepsilon k\} \cdot f(A \setminus A')/k$. *For* $j > c$, *it holds that* $\Delta\left(T_{j,\lambda_j^*} \mid A_{j-1}\right) \geq \frac{1-\varepsilon}{1+\varepsilon}|T_{j,\lambda_j^*}| \cdot f(A \setminus A')/k$.

From Claim 1,

$$f(A)-f(A\backslash A') = \Delta\left(T_{c,\lambda_c^*} \mid A\backslash A'\right) + \sum_{j=c+1}^{\ell} \Delta\left(T_{j,\lambda_j^*} \mid A_{j-1}\right) \geq \frac{(1-\varepsilon)(1-2\varepsilon)}{1+\varepsilon}\cdot f(A\backslash A'), \quad (2)$$

where Inequality 2 is proven in Appendix C.4. □

In the remainder of this subsection, we will show that a $(\beta\varepsilon)$-fraction of $V$ is discarded at each iteration $j$ of the outer **for** loop with probability at least $1/2$, where $\beta$ is a constant in terms of $\varepsilon$ as defined on Line 4 in the pseudocode. The remainder of the proofs in this section are implicitly conditioned on the behavior of the algorithm prior to iteration $j$. The next lemma describes the behavior of the number of elements that will be filtered at iteration $j+1$. Observe that the set $S_i$ defined in the next lemma is the set of elements that would be filtered at the next iteration if prefix $T_i$ is added to $A$.

**Lemma 3.** *Let $S_i = \{x \in V : \Delta\left(x \mid A \cup T_i\right) < f(A \cup T_i)/k\}$. It holds that $|S_0| = 0$, $|S_{|V|}| = |V|$, and $|S_i| \leq |S_{i+1}|$.*

By Lemma 3, we know the number of elements in $S_i$ increases from 0 to $|V|$ with $i$. Therefore, there exists a $t$ such that $t = \min\{i \in \mathbb{N} : |S_i| \geq \beta\varepsilon|V|\}$. If $\lambda^* \geq t$, $|S_{\lambda^*}| \geq \beta\varepsilon|V|$, and we will successfully filter out more than $(\beta\varepsilon)$-fraction of $V$ at the next iteration. In this case, we say that the iteration $j$ *succeeds*. Otherwise, if $\lambda^* < t$, the iteration may fail. The remainder of the proof bounds the probability that $\lambda^* < t$, which is an upper bound on the probability that iteration $j$ fails. Let $\lambda_t = \max\{\lambda \in \Lambda : \lambda < t\}$, and let $\lambda_t' = \max(\{\lambda' \in \Lambda : \sum_{\lambda \in \Lambda, \lambda' \leq \lambda \leq \lambda_t} |T_\lambda| \geq k\} \cup \{1\})$. If $\lambda^* < t$, there must be at least one index $\lambda$ between $\lambda_t'$ and $\lambda_t$ such that the block $T_\lambda'$ is bad. The next lemma bounds the probability that any block $T_\lambda'$, with $\lambda < \lambda_t$, is bad.

**Lemma 4.** *Let $t = \min\{i \in \mathbb{N} : |S_i| \geq \beta\varepsilon|V|\}$; $\lambda_t = \max\{\lambda \in \Lambda : \lambda < t\}$; $(Y_i)$ be a sequence of independent and identically distributed Bernoulli trials, where the success probability is $\beta\varepsilon$. Then for any $\lambda < \lambda_t$, $Pr\left(B[\lambda] = \textbf{false}\right) \leq Pr\left(\sum_{i=1}^{|T_\lambda'|} Y_i > \varepsilon|T_\lambda'|\right)$.*

Finally, we bound the probability that an iteration $j$ of the outer **for** loop fails. Let $B_1 = \{\lambda \in \Lambda : \lambda \leq k \text{ and } \lambda < \lambda_t\}$, $B_2 = \{\lambda \in \Lambda : |\Lambda \cap [\lambda, \lambda_t]| \leq \lceil 1/\varepsilon \rceil\}$. Then

$$Pr\,(\text{iteration } j \text{ fails}) \leq Pr\,(\exists \lambda \in B_1 \cup B_2 \text{ with } B[\lambda] = \textbf{false}) \leq 1/2, \quad (3)$$

where the proof of Inequality 3 is in Appendix C.7.

## 2.2 Proof of Theorem 1

From Section 2.1, the probability at any iteration of the outer **for** loop of successful filtering of an $(\beta\varepsilon)$-fraction of $V$ is at least $1/2$. We can model the success of the iterations as a sequence of dependent Bernoulli random variables, with success probability that depends on the results of previous trials but is always at least $1/2$.

**Success Probability of LINEARSEQ.** If there are at least $m = \lceil \log_{1-\beta\varepsilon}(1/n) \rceil$ successful iterations, the algorithm LINEARSEQ will succeed. The number of successful iterations $X_\ell$ up to and including the $\ell$-th iteration is a sum of dependent Bernoulli random variables. With some work (Lemma 6 in Appendix A), the Chernoff bounds can be applied to ensure the algorithm succeeds with probability at least $1 - 1/n$, as shown in Appendix C.2.

**Adaptivity and Query Complexity.** Oracle queries are made on Lines 6 and 13 of LINEARSEQ. The filtering on Line 6 is in one adaptive round, and the inner **for** loop is also in one adaptive round. Thus, the adaptivity is proportional to the number of iterations of the outer **for** loop, $O\left(\ell\right) = O\left(\log(n)/\varepsilon^3\right)$. For the query complexity, let $Y_i$ be the number of iterations between the $(i-1)$-th and $i$-th successful iterations of the outer **for** loop. By Lemma 6 in Appendix A, $\mathbb{E}[Y_i] \leq 2$. From here, we show in Appendix C.8 that there are at most $O\left(n/\varepsilon^3\right)$ queries in expectation.

**Approximation Ratio.** Suppose LINEARSEQ terminates successfully. We have the approximation ratio as follows:

$$f(A') \overset{(a)}{\geq} f(A) - f(A\backslash A') \overset{(b)}{\geq} f(A) - \frac{1+\varepsilon}{2(1-\varepsilon+\varepsilon^2)}f(A) \overset{(c)}{\geq} \frac{1}{4 + \frac{4(2-\varepsilon)}{(1-\varepsilon)(1-2\varepsilon)}\cdot\varepsilon}f(O),$$

where Inequality (a) is from submodularity of $f$, Inequality (b) is from Lemma 2, and Inequality (c) is from Lemma 1.

## 3 Improving to Nearly the Optimal Ratio

In this section, we describe how to obtain the nearly optimal ratio in nearly optimal query and adaptive complexities (Section 3.2). First, in Section 3.1, we describe THRESHOLDSEQ, a parallelizable procedure to add all elements with gain above a constant threshold to the solution. In Section 3.2, we describe PARALLELGREEDYBOOST and finally the main algorithm LS+PGB. Because of space constraints, the algorithms are described in the main text at a high level only, with detailed descriptions and proofs deferred to Appendices D and E.

### 3.1 The THRESHOLDSEQ Procedure

In this section, we discuss the algorithm THRESHOLDSEQ, which adds all elements with gain above an input threshold $\tau$ up to accuracy $\varepsilon$ in $O(\log n)$ adaptive rounds and $O(n)$ queries in expectation. Pseudocode is given in Alg. 4 in Appendix D.

**Overview.** The goal of this algorithm is, given an input threshold $\tau$ and size constraint $k$, to produce a set of size at most $k$ such that the average gain of elements added is at least $\tau$. As discussed in Section 1, this task is an important subroutine of many algorithms for submodular optimization (including our final algorithm), although by itself it does not produce any approximation ratio for SM. The overall strategy of our parallelizable algorithm THRESHOLDSEQ is analogous to that of LINEARSEQ, although THRESHOLDSEQ is considerably simpler to analyze. The following theorem summarizes the theoretical guarantees of THRESHOLDSEQ and the proofs are in Appendix D.

**Theorem 2.** *Suppose* THRESHOLDSEQ *is run with input* $(f, k, \varepsilon, \delta, \tau)$*. Then, the algorithm has adaptive complexity* $O(\log(n/\delta)/\varepsilon)$ *and outputs* $A \subseteq \mathcal{N}$ *with* $|A| \leq k$ *such that the following properties hold: 1) The algorithm succeeds with probability at least* $1 - \delta/n$*. 2) There are* $O(n/\varepsilon)$ *oracle queries in expectation. 3) It holds that* $f(A)/|A| \geq (1 - \varepsilon)\tau/(1 + \varepsilon)$*. 4) If* $|A| < k$*, then* $\Delta(x \mid A) < \tau$ *for all* $x \in \mathcal{N}$*.*

### 3.2 The PARALLELGREEDYBOOST Procedure and the Main Algorithm

In this section, we describe the greedy algorithm PARALLEL-GREEDYBOOST (PGB, Alg. 2) that uses multiple calls to THRESH-OLDSEQ with descending thresholds. Next, our state-of-the-art algorithm LS+PGB is specified.

**Description of PARALLEL-GREEDYBOOST.** This procedure takes as input the results from running an $\alpha$-approximation algorithm on the instance $(f, k)$ of SM; thus, PARALLELGREEDYBOOST is not meant to be used as a

---

**Algorithm 2** The PARALLELGREEDYBOOST procedure.

1: **Input:** evaluation oracle $f : 2^{\mathcal{N}} \rightarrow \mathbb{R}^+$, constraint $k$, constant $\alpha$, value $\Gamma$ such that $\Gamma \leq f(O) \leq \Gamma/\alpha$, accuracy parameter $\varepsilon$
2: Initialize $\tau \leftarrow \Gamma/(\alpha k)$, $\delta \leftarrow 1/(\log_{1-\varepsilon}(\alpha/3) + 1)$, $A \leftarrow \emptyset$
3: **while** $\tau \geq \Gamma/(3k)$ **do**
4:     $\tau \leftarrow \tau(1 - \varepsilon)$
5:     $S \leftarrow$ THRESHOLDSEQ$(f_A, \mathcal{N}, k - |A|, \delta, \varepsilon/3, \tau)$
6:     $A \leftarrow A \cup S$
7:     **if** $|A| = k$ **then**
8:         **return** $A$
9: **return** $A$

---

standalone algorithm. Namely, PARALLELGREEDYBOOST takes as input $\Gamma$, the solution value of an $\alpha$-approximation algorithm for SM; this solution value $\Gamma$ is then boosted to ensure the ratio $1 - 1/e - \varepsilon$ on the instance. The values of $\Gamma$ and $\alpha$ are used to produce an initial threshold value $\tau$ for THRESHOLDSEQ. Then, the threshold value is iteratively decreased by a factor of $(1 - \varepsilon)$ and the call to THRESHOLDSEQ is iteratively repeated to build up a solution, until a minimum value for the threshold of $\Gamma/(3k)$ is reached. Therefore, THRESHOLDSEQ is called at most $O(\log(1/\alpha)/\varepsilon)$ times. We remark that $\alpha$ is not required to be a constant approximation ratio.

**Theorem 3.** *Let* $(f, k)$ *be an instance of* SM*. Suppose an* $\alpha$*- approximation algorithm for* SM *is used to obtain* $\Gamma$*, where the approximation ratio* $\alpha$ *holds with probability* $1 - p_\alpha$*. For any constant* $\varepsilon > 0$*, the algorithm* PARALLELGREEDYBOOST *has adaptive complexity* $O\left(\frac{\log \alpha^{-1}}{\varepsilon^2} \log\left(\frac{n \log(\alpha^{-1})}{\varepsilon}\right)\right)$

*and outputs $A \in \mathcal{N}$ with $|A| \le k$ such that the following properties hold: 1) The algorithm succeeds with probability at least $1 - 1/n - p_\alpha$. 2) If the algorithm succeeds, there are $O\left(n \log\left(\alpha^{-1}\right)/\varepsilon^2\right)$ oracle queries in expectation. 3) If the algorithm succeeds, $f(A) \ge (1 - 1/e - \varepsilon)f(O)$, where $O$ is an optimal solution to the instance $(f, k)$.*

*Proof.* **Success Probability.** For the **while** loop in Line 3-8, there are no more than $\lceil \log_{1-\varepsilon}(\alpha/3) \rceil$ iterations. If THRESHOLDSEQ completes successfully at every iteration, Algorithm 2 also succeeds. The probability that this occurs is lower bounded in Appendix E.1.1. For the remainder of the proof of Theorem 3, we assume that every call to THRESHOLDSEQ succeeds.

**Adaptive and Query Complexity.** There are at most $\lceil \log_{1-\varepsilon}(\alpha/3) \rceil$ iterations of the **while** loop. Since $\log(x) \le x - 1$, $\lceil \log_{1-\varepsilon}(\alpha/3) \rceil = \lceil \frac{\log(\alpha/3)}{\log(1-\varepsilon)} \rceil$, and $\varepsilon < 1 - 1/e$, it holds that $\lceil \log_{1-\varepsilon}(\alpha/3) \rceil \le \lceil \frac{\log(3/\alpha)}{\varepsilon} \rceil$. And for each iteration, queries to the oracle happen only on Line 5, the call to THRESHOLDSEQ. Since the adaptive and query complexity of THRESHOLDSEQ is $O\left(\log(n/\delta)/\varepsilon\right)$ and $O\left(n/\varepsilon\right)$, the adaptive and query complexities for Algorithm 2 are $O\left(\frac{\log \alpha^{-1}}{\varepsilon^2} \log\left(\frac{n \log\left(\alpha^{-1}\right)}{\varepsilon}\right)\right), O\left(\frac{\log \alpha^{-1}}{\varepsilon^2} n\right)$, respectively.

**Approximation Ratio.** Let $A_j$ be the set $A$ we get after Line 6, and let $S_j$ be the set returned by THRESHOLDSEQ in iteration $j$ of the **while** loop. Let $\ell$ be the number of iterations of the **while** loop.

First, in the case that $|A| < k$ at termination, THRESHOLDSEQ returns $0 \le |S_\ell| < k - |A_{\ell-1}|$ at the last iteration. From Theorem 2, for any $o \in O$, $\Delta\left(o \,|\, A\right) < \tau < \Gamma/(3k)$. By submodularity and monotonicity, $f(O) - f(A) \le f(O \cup A) - f(A) \le \sum_{o \in O \setminus A} \Delta\left(o \,|\, A\right) \le \sum_{o \in O \setminus A} \Gamma/(3k) \le f(O)/3$, and the ratio holds.

Second, consider the case that $|A| = k$. Suppose in iteration $j + 1$, THRESHOLDSEQ returns a nonempty set $S_{j+1}$. Then, in the previous iteration $j$, THRESHOLDSEQ returns a set $S_j$ that $0 \le |S_j| < k - |A_{j-1}|$. From Theorem 2,

$$f(O) - f(A_{j+1}) \le \left(1 - \frac{(1 - \varepsilon/3)(1 - \varepsilon)}{(1 + \varepsilon/3)k}|A_{j+1} \setminus A_j|\right)(f(O) - f(A_j)). \tag{4}$$

The above inequality also holds when $A_{j+1} = A_j$. Therefore, it holds that

$$f(O) - f(A) \le e^{-\frac{(1-\varepsilon/3)(1-\varepsilon)}{1+\varepsilon/3}}f(O) \le (1/e + \varepsilon)f(O). \tag{5}$$

The detailed proof of Inequality 4 and 5 can be found in Appendix E.1.2. $\qquad\square$

**Main Algorithm: LS+PGB.** To obtain the main algorithm of this paper (and its nearly optimal theoretical guarantees), we use PARALLELGREEDYBOOST with the solution value $\Gamma$ and ratio $\alpha$ given by LINEARSEQ. Because this choice requires an initial run of LINEARSEQ, we denote this algorithm by LS+PGB. Thus, LS+PGB integrates LINEARSEQ and PARALLELGREEDYBOOST to get nearly the optimal $1 - 1/e$ ratio with query complexity of $O\left(n\right)$ and adaptivity of $O\left(\log(n)\right)$.

## 4 Empirical Evaluation

In this section, we demonstrate that the empirical performance of LS+PGB outperforms that of FAST for the metrics of total time, total queries, adaptive rounds, and objective value across six applications of SM: maximum cover on random graphs (MaxCover), twitter feed summarization (TweetSumm), image summarization (ImageSumm), influence maximization (Influence), revenue maximization (RevMax), and Traffic Speeding Sensor Placement (Traffic). See Appendix H.2 for the definition of the objectives. The sizes $n$ of the ground sets range from $n = 1885$ to $100000$.

**Implementation and Environment.** We evaluate the same implementation of FAST used in Breuer et al. [9]. Our implementation of LS+PGB is parallelized using the Message Passing Interface (MPI) within the same Python codebase as FAST (see the Supplementary Material for source code). Practical optimizations to LINEARSEQ are made, which do not compromise the theoretical guarantees, which are discussed in Appendix G. The hardware of the system consists of 40 Intel(R) Xeon(R) Gold 5218R CPU @ 2.10GHz cores (with 80 threads available), of which up to 75 threads are

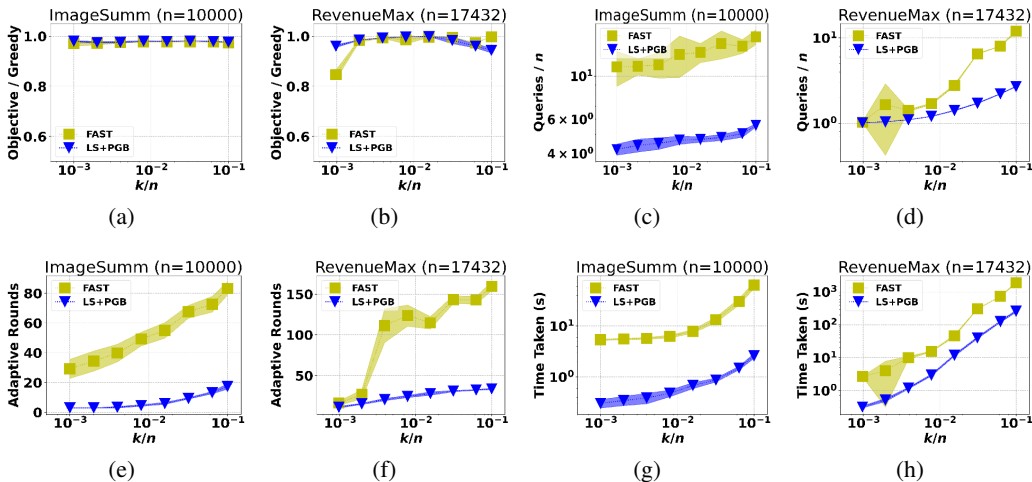

Figure 2: Evaluation of adaptive algorithms on ImageSumm and RevenueMax in terms of objective value normalized by the standard greedy value (Figure 2(a) - 2(b)), total number of queries (Figure 2(c) - 2(d)), total adaptive rounds (Figure 2(e) - 2(f)) and the time required by each algorithm (Figure 2(g) - 2(h))

made available to the algorithms for the experiments. On each instance, the algorithms are repeated independently for five repetitions, and the mean and standard deviation of the objective value, total queries, adaptive rounds and parallel (wall clock) runtime to the submodular function is plotted.

**Parameters.** The parameters $\varepsilon, \delta$ of FAST are set to enforce the nominal ratio of $1 - 1/e - 0.1 \approx 0.53$ with probability $0.95$; these are the same parameter settings for FAST as in the Breuer et al. [9] evaluation. The $\varepsilon$ parameter of LS+PGB is set to enforce the same ratio with probability $1 - 2/n$. With these parameters, FAST ensures its ratio only if $k \geq \theta(\varepsilon, \delta, k) = 2 \log(2\delta^{-1} \log(\frac{1}{\varepsilon} \log(k)))/\varepsilon^2 (1 - 5\varepsilon) \geq 7103$. Since $k < 7103$ on many of our instances, FAST is evaluated in these instances as a theoretically motivated heuristic. In contrast, the ratio of LS+PGB holds on all instances evaluated. We use exponentially increasing $k$ values from $n/1000$ to $n/10$ for each application to explore the behavior of each algorithm across a broad range of instance sizes.

**Overview of Results.** Figure 2 illustrates the comparison with FAST across the ImageSumm and RevenueMax application; results on other applications are shown in Appendix H. **Runtime:** LS+PGB is faster than FAST by more than $1\%$ on $80\%$ of instances evaluated; and is faster by an order of magnitude on $14\%$ of instances. **Objective value:** LS+PGB achieves higher objective by more than $1\%$ on $50\%$ of instances, whereas FAST achieves higher objective by more than $1\%$ on $8\%$ of instances. **Adaptive rounds:** LS+PGB achieves more than $1\%$ fewer adaptive rounds on $75\%$ of instances, while FAST achieves more than $1\%$ fewer adaptive rounds on $22\%$ of instances. **Total queries:** LS+PGB uses more than $1\%$ fewer queries on $84\%$ of scenarios with FAST using more than $1\%$ fewer queries on $9\%$ of scenarios. In summary, LS+PGB frequently gives substantial improvement in objective value, queries, adaptive rounds, and parallel runtime. Comparison of the arithmetic means of the metrics over all instances is given in Table 2. Finally, FAST and LS+PGB show very similar linear speedup with the number of processors employed: as shown in Fig. 7.

## 5 Concluding Remarks

In this work, we have introduced the algorithm LS+PGB, which is highly parallelizable and achieves state-of-the-art empirical performance over any previous algorithm for SM; also, LS+PGB is nearly optimal theoretically in terms of query complexity, adaptivity, and approximation ratio. An integral component of LS+PGB is our preprocessing algorithm LINEARSEQ, which reduces the interval containing OPT to a small constant size in expected linear time and low adaptivity, which may be independently useful. Another component of LS+PGB is the THRESHOLDSEQ procedure, which adds all elements with gain above a threshold in a parallelizable manner and improves existing algorithms in the literature for the same task.

## Acknowledgements

The work of Yixin Chen, Tonmoy Dey, and Alan Kuhnle was partially supported by Florida State University. The authors have received no third-party funding in direct support of this work. The authors have no additional revenues from other sources related to this work.

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
