# A Probability Lemma and Concentration Bounds

In this section, we state the Chernoff bound and prove a useful lemma (Lemma 6) for working with a sequence of dependent Bernoulli trials. Lemma 6 is applied in the analysis of both THRESHOLDSEQ and LINEARSEQ.

**Lemma 5.** *(Chernoff bounds [31]). Suppose $X_1, \dots, X_n$ are independent binary random variables such that $Pr(X_i = 1) = p_i$. Let $\mu = \sum_{i=1}^{n} p_i$, and $X = \sum_{i=1}^{n} X_i$. Then for any $\delta \geq 0$, we have*

$$Pr(X \geq (1 + \delta)\mu) \leq e^{-\frac{\delta^2 \mu}{2+\delta}}. \tag{6}$$

*Moreover, for any $0 \leq \delta \leq 1$, we have*

$$Pr(X \leq (1 - \delta)\mu) \leq e^{-\frac{\delta^2 \mu}{2}}. \tag{7}$$

**Lemma 6.** *Suppose there is a sequence of $n$ Bernoulli trials: $X_1, X_2, \dots, X_n$, where the success probability of $X_i$ depends on the results of the preceding trials $X_1, \dots, X_{i-1}$. Suppose it holds that*

$$Pr(X_i = 1 | X_1 = x_1, X_2 = x_2, \dots, X_{i-1} = x_{i-1}) \geq \eta,$$

*where $\eta > 0$ is a constant and $x_1, \dots, x_{i-1}$ are arbitrary.*

*Then, if $Y_1, \dots, Y_n$ are independent Bernoulli trials, each with probability $\eta$ of success, then*

$$Pr\left(\sum_{i=1}^{n} X_i \leq b\right) \leq Pr\left(\sum_{i=1}^{n} Y_i \leq b\right),$$

*where $b$ is an arbitrary integer.*

*Moreover, let $A$ be the first occurrence of success in sequence $X_i$. Then,*

$$\mathbb{E}[A] \leq 1/\eta.$$

*Proof.* Let $Z_j = \sum_{i=1}^{j} X_i + \sum_{i=j+1}^{n} Y_i$, where $Z_0 = \sum_{i=1}^{n} Y_i$ and $Z_n = \sum_{i=1}^{n} X_i$. Our goal is to prove that $Pr(Z_n \leq b) \leq Pr(Z_0 \leq b)$. This lemma holds, if for any $j = 1, \dots, n$, $Pr(Z_j \leq b) \leq Pr(Z_{j-1} \leq b)$. With Bayes Theorem and Total Probability Theorem,

$$
\begin{aligned}
Pr(Z_j \leq b) &= Pr(X_j = 0, Z_j - X_j \leq b - 1) + Pr(X_j = 1, Z_j - X_j \leq b - 1) \\
&\quad + Pr(X_j = 0, Z_j - X_j = b) \\
&= Pr(Z_j - X_j \leq b - 1) + \sum_{Z_j - X_j = b} Pr(X_j = 0 | X_1, \dots, X_{j-1}, Y_{j+1}, \dots, Y_n) \\
&\quad \cdot Pr(X_1, \dots, X_{j-1}, Y_{j+1}, \dots, Y_n) \\
&\leq Pr(Z_{j-1} - Y_j \leq b - 1) \\
&\quad + \sum_{Z_{j-1} - Y_j = b} Pr(Y_j = 0) \cdot Pr(X_1, \dots, X_{j-1}, Y_{j+1}, \dots, Y_n) \tag{8} \\
&= Pr(Z_{j-1} - Y_j \leq b - 1) + Pr(Y_j = 0, Z_{j-1} - Y_j = b) \\
&= Pr(Z_{j-1} \leq b),
\end{aligned}
$$

where inequality 8 follows from $Pr(X_j = 0 | X_1 = x_1, X_2 = x_2, \dots, X_{j-1} = x_{j-1}) \leq 1 - \eta = Pr(Y_j = 0)$ and $Z_j - X_j = Z_{j-1} - Y_j$.

Following the first inequality, we can prove the second one as follows:

$$
\begin{aligned}
\mathbb{E}[A] &= \sum_{a \geq 1} a Pr(A = a) \\
&= \sum_{a \geq 1} a Pr(X_1 = \dots = X_{a-1} = 0, X_a = 1) \\
&= \sum_{a \geq 1} a \left(Pr(X_1 = \dots = X_{a-1} = 0) - Pr(X_1 = \dots = X_a = 0)\right)
\end{aligned}
$$

$$= 1 + \sum_{a \geq 1} Pr\left(X_1 = \ldots = X_a = 0\right)$$

$$= 1 + \sum_{a \geq 1} Pr\left(\sum_{i=1}^{a} X_a \leq 0\right)$$

$$\leq 1 + \sum_{a \geq 1} Pr\left(\sum_{i=1}^{a} Y_a \leq 0\right)$$

$$= 1 + \sum_{a \geq 1} Pr\left(Y_1 = \ldots = Y_a = 0\right)$$

$$= 1 + \sum_{a \geq 1} (1 - \eta)^a$$

$$= 1/\eta.$$

$\square$

**Lemma 7.** *Suppose there is a sequence of $n + 1$ Bernoulli trials: $X_1, X_2, \ldots, X_{n+1}$, where the success probability of $X_i$ depends on the results of the preceding trials $X_1, \ldots, X_{i-1}$, and it decreases from 1 to 0. Let $t$ be a random variable based on the $n + 1$ Bernoulli trials. Suppose it holds that*

$$Pr\left(X_i = 1 | X_1 = x_1, X_2 = x_2, \ldots, X_{i-1} = x_{i-1}, i \leq t\right) \geq \eta,$$

*where $x_1, \ldots, x_{i-1}$ are arbitrary and $0 < \eta < 1$ is a constant. Then, if $Y_1, \ldots, Y_{n+1}$ are independent Bernoulli trials, each with probability $\eta$ of success, then*

$$Pr\left(\sum_{i=1}^{t} X_i \leq bt\right) \leq Pr\left(\sum_{i=1}^{t} Y_i \leq bt\right),$$

*where $b$ is an arbitrary integer.*

*Proof.* Let $Z_j = \sum_{i=1}^{j} X_i \cdot 1_{\{i \leq t\}} + \sum_{i=j+1}^{n+1} Y_i \cdot 1_{\{i \leq t\}}$, where

$$Z_0 = \sum_{i=1}^{n+1} Y_i \cdot 1_{\{i \leq t\}} = \sum_{i=1}^{t} Y_i,$$

and

$$Z_{n+1} = \sum_{i=1}^{n+1} X_i \cdot 1_{\{i \leq t\}} = \sum_{i=1}^{t} X_i.$$

If for any $1 \leq j \leq n + 1$,

$$Pr\left(Z_j \leq bt\right) \leq Pr\left(Z_{j-1} \leq bt\right), \tag{9}$$

then,

$$Pr\left(Z_{n+1} \leq bt\right) \leq Pr\left(Z_0 \leq bt\right).$$

We prove Inequality 9 as follows,

$$Pr\left(Z_j \leq bt\right)$$
$$= Pr\left(X_j \cdot 1_{\{j \leq t\}} = 0, Z_j - X_j \cdot 1_{\{j \leq t\}} \leq bt - 1\right)$$
$$\quad + Pr\left(X_j \cdot 1_{\{j \leq t\}} = 1, Z_j - X_j \cdot 1_{\{j \leq t\}} \leq bt - 1\right)$$
$$\quad + Pr\left(X_j \cdot 1_{\{j \leq t\}} = 0, Z_j - X_j \cdot 1_{\{j \leq t\}} = bt\right)$$
$$= Pr\left(Z_j - X_j \cdot 1_{\{j \leq t\}} \leq bt - 1\right)$$
$$\quad + Pr\left(1_{\{j \leq t\}} = 0, Z_j - X_j \cdot 1_{\{j \leq t\}} = bt\right)$$
$$\quad + Pr\left(X_j = 0, 1_{\{j \leq t\}} = 1, Z_j - X_j \cdot 1_{\{j \leq t\}} = bt\right)$$
$$= Pr\left(Z_j - X_j \cdot 1_{\{j \leq t\}} \leq bt - 1\right)$$
$$\quad + Pr\left(1_{\{j \leq t\}} = 0, Z_j - X_j \cdot 1_{\{j \leq t\}} = bt\right)$$

---

**Algorithm 3** Highly Adaptive Linear-Time Algorithm

---

1: **Input:** evaluation oracle $f : 2^{\mathcal{N}} \to \mathbb{R}^+$, constraint $k$
2: Initialize $A \leftarrow \emptyset$
3: **for** $u \in \mathcal{N}$ **do**
4:     **if** $\Delta(u \mid A) \geq f(A)/k$ **then**
5:         $A \leftarrow A \cup \{u\}$
6: **return** $A' \leftarrow \{\text{last } k \text{ elements added to } A\}$

---

$$
\begin{aligned}
&+ \sum_{Z_j - X_j \cdot 1_{\{j \leq t\}} = bt, j \leq t} Pr\left(X_j = 0 | X_1, \cdots, X_{j-1}, Y_{j+1}, \cdots, Y_{n+1}, j \leq t\right) \\
&\quad \cdot Pr\left(X_1, \cdots, X_{j-1}, Y_{j+1}, \cdots, Y_{n+1}, j \leq t\right) \\
&\leq Pr\left(Z_{j-1} - Y_j \cdot 1_{\{j \leq t\}} \leq bt - 1\right) \\
&\quad + Pr\left(1_{\{j \leq t\}} = 0, Z_{j-1} - Y_j \cdot 1_{\{j \leq t\}} = bt\right) \\
&\quad + \sum_{Z_{j-1} - Y_j \cdot 1_{\{j \leq t\}} = bt, j \leq t} Pr\left(Y_j = 0\right) \cdot Pr\left(X_1, \cdots, X_{j-1}, Y_{j+1}, \cdots, Y_{n+1}, j \leq t\right) \quad (10) \\
&= Pr\left(Z_{j-1} - Y_j \cdot 1_{\{j \leq t\}} \leq bt - 1\right) \\
&\quad + Pr\left(1_{\{j \leq t\}} = 0, Z_{j-1} - Y_j \cdot 1_{\{j \leq t\}} = bt\right) \\
&\quad + Pr\left(Y_j = 0, 1_{\{j \leq t\}} = 1, Z_{j-1} - Y_j \cdot 1_{\{j \leq t\}} = bt\right) \\
&= Pr\left(Z_{j-1} - Y_j \cdot 1_{\{j \leq t\}} \leq bt - 1\right) \\
&\quad + Pr\left(Y_j \cdot 1_{\{j \leq t\}} = 0, Z_{j-1} - Y_j \cdot 1_{\{j \leq t\}} = bt\right) \\
&= Pr\left(Z_{j-1} \leq bt\right),
\end{aligned}
$$

where Inequality 10 follows from $Pr\left(X_i = 1 | X_1 = x_1, X_2 = x_2, \ldots, X_{i-1} = x_{i-1}, i \leq t\right) \geq \eta$ and $Z_j - X_j \cdot 1_{\{j \leq t\}} = Z_{j-1} - Y_j \cdot 1_{\{j \leq t\}}$. $\qquad \square$

**Lemma 8.** *(Wald's Equation [34]). Let $(X_n)_{n \in \mathbb{N}}$ be an infinite sequence of real-valued random variables and let $N$ be a nonnegative integer-valued random variable. Assume that: 1) $(X_n)_{n \in \mathbb{N}}$ are all integrable (finite-mean) random variables, 2) $\mathrm{E}\left[X_n 1_{\{N \geq n\}}\right] = \mathrm{E}\left[X_n\right] \mathrm{P}(N \geq n)$ for every natural number $n$, 3) the infinite series satisfies $\sum_{n=1}^{\infty} \mathrm{E}\left[|X_n| \, 1_{\{N \geq n\}}\right] < \infty$. Then the random sums $S_N = \sum_{n=1}^{N} X_n$ and $T_N = \sum_{n=1}^{N} \mathrm{E}\left[X_n\right]$ are integrable and $\mathrm{E}\left[S_N\right] = \mathrm{E}\left[T_N\right]$.*

## B    Highly Adaptive $0.25$-Approximation

In this section, we show that Alg. 3 achieves approximation ratio of $1/4$ in $n$ adaptive queries to the objective function $f$. Alg. 3 influences the design of LINEARSEQ, which uses similar ideas to obtain its constant ratio. See the discussion in Section 2.

**Description of Alg. 3.** This algorithm operates in one **for** loop through the ground set. Each element $u$ is added to a set $A$ iff. $\Delta(u \mid A) \geq f(A)/k$. The solution returned is the set $A'$ of the last $k$ elements added to $A$.

**Theorem 4.** *Let $O$ be an optimal solution to SM on instance $(f, k)$. Then the set $A'$ produced after running Alg. 3 on this instance satisfies $4f(A') \geq f(O)$.*

*Proof.*

**Lemma 9.** *At termination of Alg. 3, $f(O) \leq 2f(A)$.*

*Proof.* For each $o \in O \backslash A$, let $j(o) + 1$ be the iteration of the **for** loop in which $o$ is processed, and $A_{j(o)}$ be $A$ after iteration $j(o)$. Thus, $\Delta\left(o \mid A_{j(o)}\right)) < f(A_{j(o)})/k$. Then

$$
f(O) - f(A) \leq f(O \cup A) - f(A) \tag{11}
$$

$$
\leq \sum_{o \in O \backslash A} \Delta(o \mid A) \tag{12}
$$

$$\leq \sum_{o \in O \setminus A} \Delta\left(o \mid A_{j(o)}\right) \tag{13}$$

$$\leq \sum_{o \in O \setminus A} f(A_{j(o)})/k$$

$$\leq f(A), \tag{14}$$

where Inequalities 12 and 13 follow from submodularity, and Inequalities 11 and 14 follow from monotonicity. □

**Lemma 10.** *At termination of Alg. 3, $f(A) \leq 2f(A')$.*

*Proof.* If $|A| < k$, then $A' = A$ and there is nothing to show. So suppose $|A| \geq k$. Let $A'_i = \{a_1, \ldots, a_{i-1}\}$.

$$f(A) - f(A \setminus A') = \sum_{i=1}^{k} \Delta\left(a_i \mid A \setminus A' \cup A'_i\right)$$

$$\geq \sum_{i=1}^{k} f\left((A \setminus A') \cup A'_i\right)/k$$

$$\geq \sum_{i=1}^{k} f\left(A \setminus A'\right)/k$$

$$= f\left(A \setminus A'\right).$$

Therefore, $f\left(A \setminus A'\right) \leq f(A)/2$. Now,

$$f(A') \geq f(A) - f(A \setminus A') \tag{15}$$

$$\geq f(A) - f(A)/2 = f(A)/2,$$

where Inequality 15 is by submodularity of $f$. □

By Lemma 9 and 10, we have

$$f(O) \leq 2f(A) \leq 4f(A').$$

□

# C  Analysis of LINEARSEQ

## C.1  Proof of Lemma 1

**Lemma 1.** *At successful termination of LINEARSEQ, $f(O) \leq 2f(A)$, where $O \subseteq \mathcal{N}$ is an optimal solution of size $k$.*

*Proof.* For each $o \in O \setminus A$, let $j(o) + 1$ be the iteration where $o$ is filtered out, and $A_{j(o)}$ be $A$ after iteration $j(o)$. Thus, $\Delta\left(o \mid A_{j(o)}\right)) < f(A_{j(o)})/k$. Then

$$f(O) - f(A) \leq f(O \cup A) - f(A) \tag{16}$$

$$\leq \sum_{o \in O \setminus A} \Delta\left(o \mid A\right) \tag{17}$$

$$\leq \sum_{o \in O \setminus A} \Delta\left(o \mid A_{j(o)}\right) \tag{18}$$

$$\leq \sum_{o \in O \setminus A} f(A_{j(o)})/k$$

$$\leq f(A), \tag{19}$$

where Inequalities 17 and 18 follow from submodularity, and Inequalities 16 and 19 follow from monotonicity. □

## C.2 Probability LINEARSEQ is Successful

*Proof.* Let $Y_\ell$ be the number of successes in $\ell$ independent Bernoulli random variables with success probability 1/2. Then,

$$Pr\left(\text{Algorithm 1 succeeds}\right) \geq Pr\left(Y_\ell \geq m\right) \tag{20}$$
$$\geq Pr\left(Y_\ell \geq \log(n)/(\beta\varepsilon)\right)$$
$$= 1 - Pr\left(Y_\ell \leq \log(n)/(\beta\varepsilon)\right)$$
$$\geq 1 - e^{-\frac{1}{2}\left(\frac{2\beta\varepsilon+1}{2(\beta\varepsilon+1)}\right)^2 \cdot 2\left(1+\frac{1}{\beta\varepsilon}\right)\log(n)} \tag{21}$$
$$= 1 - \left(\frac{1}{n}\right)^{\frac{(2\beta\varepsilon+1)^2}{4\beta\varepsilon(\beta\varepsilon+1)}}$$
$$\geq 1 - \frac{1}{n},$$

where Inequalities 20 and 21 follow from Lemma 6 and Lemma 5, respectively. $\square$

## C.3 Proof of Claim 1

**Claim 1.** It holds that $\Delta\left(T_{c,\lambda_c^*} \mid A\backslash A'\right) \geq (1-\varepsilon)\max\{0, |T_{c,\lambda_c^*} \cap A'| - 2\varepsilon k\} \cdot f(A\backslash A')/k$. For $j > c$, it holds that $\Delta\left(T_{j,\lambda_j^*} \mid A_{j-1}\right) \geq \frac{1-\varepsilon}{1+\varepsilon}|T_{j,\lambda_j^*}| \cdot f(A\backslash A')/k$.

*Proof.* In this proof, let $T_{j,i} = (v_1, \ldots, v_i)$ be the prefix of $V$ of length $i$ at iteration $j$. Likewise, define $T'_{j,i} = T_{j,i} \setminus T_{j,i-1}$. Say block $T'_{j,k}$ is *bad* if this block does not satisfy the condition in Line 13 during iteration $j$.

First, consider the case that $j > c$. It holds that $T_{j,\lambda_j^*} \subseteq A'$. Thus, $|T_{j,\lambda_j^*}| \leq k$. If $|T_{j,\lambda_j^*}| = 0$, the result holds. If $0 < |T_{j,\lambda_j^*}| \leq k$, then there are several blocks in $T_{j,\lambda_j^*}$, where the last one is bad and all the previous ones are good. Since the block size exponentially increases with ratio $1 + \varepsilon$ when $\lambda_i \leq k$, it holds that $|T'_{j,\lambda_j^*}| \leq \varepsilon/(1+\varepsilon)|T_{j,\lambda_j^*}|$. Then,

$$\Delta\left(T_{j,\lambda_j^*} \mid A_{j-1}\right) = \sum_{\lambda_i \leq \lambda_j^*} \Delta\left(T'_{j,\lambda_i} \mid A_{j-1} \cup T_{j,\lambda_{i-1}}\right)$$
$$\geq \sum_{\lambda_i < \lambda_j^*} \Delta\left(T'_{j,\lambda_i} \mid A_{j-1} \cup T_{j,\lambda_{i-1}}\right) \tag{22}$$
$$\geq \sum_{\lambda_i < \lambda_j^*} (1-\varepsilon)|T'_{j,\lambda_i}| \cdot f(A_{j-1} \cup T_{j,\lambda_{i-1}})/k \tag{23}$$
$$\geq (1-\varepsilon)|T_{j,\lambda_j^*}\backslash T'_{j,\lambda_j^*}| \cdot f(A\backslash A')/k \tag{24}$$
$$\geq \frac{1-\varepsilon}{1+\varepsilon}|T_{j,\lambda_j^*}| \cdot f(A\backslash A')/k,$$

where Inequalities 22 and 24 follow from monotonicity, and Inequality 23 follows from the property of good block.

Now consider the case $j = c$. In this case, block $T'_{c,\lambda_c^*}$ is always a bad block. And, if $|T_{c,\lambda_c^*}| \leq k$, all the previous blocks are good; if $|T_{c,\lambda_c^*}| > k$, several previous blocks are good total with at least $k$ elements. Let $\lambda_v = \min\{\lambda_v \in \Lambda : \cup_{\lambda_v \leq \lambda_i \leq \lambda_c^*} T'_{c,\lambda_i} \subseteq A'\}$. Then, block $T'_{c,\lambda_v}$ to the block before $T'_{c,\lambda_c^*}$ are all good blocks. For any $\lambda_i \in \Lambda$, it holds that $|T'_{c,\lambda_i}| \leq \varepsilon k$. Then,

$$\Delta\left(T_{c,\lambda_c^*} \mid A\backslash A'\right) = \sum_{\lambda_i \leq \lambda_c^*} \Delta\left(T'_{c,\lambda_i} \mid A\backslash A' \cup T_{c,\lambda_{i-1}}\right)$$
$$\geq \sum_{\lambda_v \leq \lambda_i < \lambda_c^*} \Delta\left(T'_{c,\lambda_i} \mid A\backslash A' \cup T_{c,\lambda_{i-1}}\right) \tag{25}$$
$$\geq (1-\varepsilon)\sum_{\lambda_v \leq \lambda_i < \lambda_c^*} |T'_{c,\lambda_i}| \cdot f(A\backslash A' \cup T_{c,\lambda_{i-1}})/k \tag{26}$$

$$\geq (1-\varepsilon) \max\left\{0, \left(|T_{c,\lambda_c^*} \cap A'| - |T'_{c,\lambda_{v-1}}| - |T'_{c,\lambda_c^*}|\right)\right\} \cdot f(A\backslash A')/k \quad (27)$$

$$\geq (1-\varepsilon) \max\{0, |T_{c,\lambda_c^*} \cap A'| - 2\varepsilon k\} \cdot f(A\backslash A')/k,$$

where Inequality 25 follows from monotonicity, Inequality 26 follows from the property of good block, Inequality 27 follows from monotonicity, $\cup_{\lambda_v \leq \lambda_i < \lambda_c^*} T'_{c,\lambda_i} \subseteq \left(T_{c,\lambda_c^*} \cap A'\right)$, and

$$\left(\left(T_{c,\lambda_c^*} \cap A'\right) \backslash \cup_{u=v}^{i-1} T'_{c,\lambda_i}\right) \subseteq \left(T'_{c,\lambda_{v-1}} \cup T'_{c,\lambda_c^*}\right).$$

$\square$

## C.4   Proof of Inequality 2

*Proof of Inequality 2.*

$$f(A) - f(A\backslash A') = \Delta\left(T_{c,\lambda_c^*} \,|\, A\backslash A'\right) + \sum_{j=c+1}^{\ell} \Delta\left(T_{j,\lambda_j^*} \,|\, A_{j-1}\right)$$

$$\geq (1-\varepsilon) \max\{0, |T_{c,\lambda_c^*} \cap A'| - 2\varepsilon k\} \cdot f(A\backslash A')/k + \frac{1-\varepsilon}{1+\varepsilon}\left(k - |T_{c,\lambda_c^*} \cap A'|\right) \cdot f(A\backslash A')/k$$

$$= (1-\varepsilon)\left(\max\left\{2\varepsilon - \frac{|T_{c,\lambda_c^*} \cap A'|}{(1+\varepsilon)k}, \frac{\varepsilon|T_{c,\lambda_c^*} \cap A'|}{(1+\varepsilon)k}\right\} + \frac{1}{1+\varepsilon} - 2\varepsilon\right) \cdot f(A\backslash A')$$

$$\geq \frac{(1-\varepsilon)(1-2\varepsilon)}{1+\varepsilon} \cdot f(A\backslash A') \quad (28)$$

where Inequality 28 follows from the fact that $\max\left\{2\varepsilon - \frac{|T_{c,\lambda_c^*} \cap A'|}{(1+\varepsilon)k}, \frac{\varepsilon|T_{c,\lambda_c^*} \cap A'|}{(1+\varepsilon)k}\right\} \geq 2\varepsilon^2/(1+\varepsilon)$. $\square$

## C.5   Proof of Lemma 3

**Lemma 3.** *Let $S_i = \{x \in V : \Delta\left(x \,|\, A \cup T_i\right) < f(A \cup T_i)/k\}$. It holds that $|S_0| = 0$, $|S_{|V|}| = |V|$, and $|S_i| \leq |S_{i+1}|$.*

*Proof.* After Line 6, for any $x \in V$, $\Delta\left(x \,|\, A\right) \geq f(A)/k$, $\Delta\left(x \,|\, V \cup A\right) = 0$. Since $T_0 = \emptyset$ and $T_{|V|} = V$,

$$\begin{aligned} |S_0| &= |\{x \in V : \Delta\left(x \,|\, A\right) < f(A)/k\}| &= 0, \\ |S_{|V|}| &= |\{x \in V : \Delta\left(x \,|\, A \cup V\right) < f(A \cup V)/k\}| = |V|. \end{aligned}$$

Due to submodularity and monotonicity, for any $x \in S_i$,

$$\Delta\left(x \,|\, A \cup T_{i+1}\right) \leq \Delta\left(x \,|\, A \cup T_i\right) < f(A \cup T_i)/k \leq f(A \cup T_{i+1})/k.$$

So, $x \in S_{i+1}$, and $S_i$ is the subset of $S_{i+1}$, which means $|S_i| \leq |S_{i+1}|$. $\square$

## C.6   Proof of Lemma 4

**Lemma 4.** *Let $t = \min\{i \in \mathbb{N} : |S_i| \geq \beta\varepsilon|V|\}$; $\lambda_t = \max\{\lambda \in \Lambda : \lambda < t\}$; $(Y_i)$ be a sequence of independent and identically distributed Bernoulli trials, where the success probability is $\beta\varepsilon$. Then for any $\lambda < \lambda_t$, $Pr\left(B[\lambda] = \textbf{false}\right) \leq Pr\left(\sum_{i=1}^{|T'_\lambda|} Y_i > \varepsilon|T'_\lambda|\right)$.*

*Proof.* Call an element $v_u \in V$ *bad* if $\Delta\left(v_u \,|\, A \cup T_{u-1}\right) < f(A \cup T_{u-1})/k$. The random permutation of $V$ can be regarded as $|V|$ dependent Bernoulli trials, where $Pr\left(v_i \text{ is bad}|v_1, \cdots, v_{i-1}\right) = |S_{i-1}|/|V|$. For block $\lambda < \lambda_t$ to fail the condition on Line 13, the number of bad elements in $T'_\lambda$ must be more than $\varepsilon|T'_\lambda|$. For $u < t$, the probability that $v_u$ is bad is less than $\beta\varepsilon$ by Lemma 3. Then,

$$Pr\left(B[\lambda] = \textbf{false}\right) \leq Pr\left(\# \text{ of bad elements in } T'_\lambda > \varepsilon|T'_\lambda|\right)$$

$$\leq Pr\left(\sum_{v_u \in T'_\lambda} 1_{\{v_u \text{ is bad}\}} > \varepsilon |T'_\lambda|\right)$$

$$\leq Pr\left(\sum_{i=1}^{|T'_\lambda|} Y_i > \varepsilon |T'_\lambda|\right), \tag{29}$$

where Inequality 29 follows from Lemma 7. □

### C.7 Proof of Inequality 3

**Lemma 11.** *Let $(Y_i)$ be a sequence of independent and identically distributed Bernoulli trials, where the success probability is $\beta\varepsilon$. Then for a constant integar $\alpha$, $Pr\left(\sum_{i=1}^{\alpha} Y_i > \varepsilon\alpha\right) \leq \min\{\beta, e^{-\frac{(1-\beta)^2}{1+\beta}\varepsilon\alpha}\}$.*

*Proof.* Since $(Y_i)$ is a sequence of i.i.d. Bernoulli trails with success probability $\beta\varepsilon$, it holds that $\mathbb{E}\left[\sum_{i=1}^{\alpha} Y_i\right] = \beta\varepsilon\alpha$. For small $\alpha$, by Markov's inequality, we can bound the probability as follows,

$$Pr\left(\sum_{i=1}^{\alpha} Y_i > \varepsilon\alpha\right) \leq \frac{\mathbb{E}\left[\sum_{i=1}^{\alpha} Y_i\right]}{\varepsilon\alpha} = \beta.$$

For large $\alpha$, there exists a tighter bound by the application of Lemma 5,

$$Pr\left(\sum_{i=1}^{\alpha} Y_i > \varepsilon\alpha\right) \leq e^{-\frac{(1-\beta)^2}{1+\beta}\varepsilon\alpha}.$$

□

*Proof of Inequality 3.* First, by Lemma 8:

$$Pr\left(\text{iteration } j \text{ fails}\right) \leq Pr\left(\exists\lambda \in B_1 \cup B_2 \text{ with } B[\lambda] = \textbf{false}\right)$$

$$= \mathbb{E}\left[1_{\{\exists\lambda \in B_1 \cup B_2 \text{ with } B[\lambda] = \textbf{false}\}}\right]$$

$$\leq \mathbb{E}\left[\sum_{\lambda \in B_1} 1_{\{B[\lambda] = \textbf{false}\}} + \sum_{\lambda \in B_2} 1_{\{B[\lambda] = \textbf{false}\}}\right]$$

$$= \mathbb{E}\left[\sum_{\lambda \in B_1} 1_{\{B[\lambda] = \textbf{false}\}}\right] + \mathbb{E}\left[\sum_{\lambda \in B_2} 1_{\{B[\lambda] = \textbf{false}\}}\right]$$

$$= \mathbb{E}\left[\sum_{\lambda \in B_1} \mathbb{E}\left[1_{\{B[\lambda] = \textbf{false}\}}\right]\right] + \mathbb{E}\left[\sum_{\lambda \in B_2} \mathbb{E}\left[1_{\{B[\lambda] = \textbf{false}\}}\right]\right], \tag{30}$$

where Equation 30 holds, since the sequence $(1_{\{B[\lambda] = \textbf{false}\}})_\lambda$ and the random variable $t$ follow the assumptions in Lemma 8: 1) $1_{\{B[\lambda] = \textbf{false}\}}$s are all integrable random variables, because they only take the value 0 and 1; 2) $t$ is a stopping time since it only depends on the previous $t-1$ selections; 3) $Pr\left(1_{\{t \geq n\}} = 0\right) = 1$ for any $n \geq |V|$.

To bound the first term of Inequality 30, we have from Lemma 4,

$$\mathbb{E}\left[\sum_{\lambda \in B_1} \mathbb{E}\left[1_{\{B[\lambda] = \textbf{false}\}}\right]\right] \leq \mathbb{E}\left[\sum_{\lambda \in B_1} Pr\left(\sum_{i=1}^{|T'_\lambda|} Y_i > \varepsilon |T'_\lambda|\right)\right]$$

$$\leq \mathbb{E}\left[\sum_{\lambda \in \{\lfloor(1+\varepsilon)^u\rfloor : u \geq 1\}} Pr\left(\sum_{i=1}^{|T'_\lambda|} Y_i > \varepsilon |T'_\lambda|\right)\right] \tag{31}$$

$$\leq \sum_{\lambda \in \{\lfloor(1+\varepsilon)^u\rfloor : u \geq 1\}} \min\{\beta, e^{-\frac{(1-\beta)^2}{1+\beta}\varepsilon |T'_\lambda|}\} \tag{32}$$

$$\leq \sum_{\lambda \geq 1} \min\{\beta, e^{-\varepsilon\lambda/2}\} \tag{33}$$

$$\leq a\beta + \sum_{\lambda=a+1}^{\infty} e^{-\varepsilon\lambda/2}, \text{ where } a = \left\lfloor \frac{1}{8\beta} \right\rfloor = \left\lfloor \frac{2}{\varepsilon} \log\left( \frac{8}{1-e^{-\varepsilon/2}} \right) \right\rfloor$$

$$\leq a\beta + \frac{e^{-\varepsilon(a+1)/2}}{1-e^{-\varepsilon/2}}$$

$$\leq \frac{1}{8} + \frac{1}{8} = \frac{1}{4},$$

where Inequality 31 follows from $B_1 \subseteq \{\lfloor (1+\varepsilon)^u \rfloor : u \geq 1\}$, and let $\{\lfloor (1+\varepsilon)^u \rfloor : u \geq 1\} = \{\lambda_1, \lambda_2, \cdots\}$, define $|T'_{\lambda_i}| = \lambda_i - \lambda_{i-1}$; Inequality 32 follows from Lemma 11; and Inequality 33 follows from $|T'_\lambda| < \lambda$ and $\beta < 1/2$.

For the second term in Inequality 30, from Lemma 4, Lemma 11, and the fact that $|B_2| < \lceil 1/\varepsilon \rceil$, we have

$$\mathbb{E}\left[ \sum_{\lambda \in B_2} \mathbb{E}\left[ 1_{\{B[\lambda]=\mathbf{false}\}} \right] \right] \leq \mathbb{E}\left[ \sum_{\lambda \in B_2} Pr\left( \sum_{i=1}^{|T'_\lambda|} Y_i > \varepsilon |T'_\lambda| \right) \right] \leq \beta \lceil 1/\varepsilon \rceil \leq \frac{1}{4}.$$

From the above two inequalities, we have that

$$Pr\,(\text{iteration } j \text{ fails}) \leq 1/2.$$

$\square$

## C.8 Query Complexity of LINEARSEQ

*Query Complexity.* Let $V_j$ be the value of $V$ after filtering on Line 6 during iteration $j$, and let $Y_i$ be the number of iterations between the $(i-1)$-th success and $i$-th success. By Lemma 6 in Appendix A, $\mathbb{E}[Y_i] \leq 2$. Observe that $|V_0| = |\mathcal{N}| = n$. Then, the expected number of queries is bounded as follows:

$$\mathbb{E}[Queries] \leq \sum_{j=1}^{\ell} \mathbb{E}\left[ |V_{j-1}| + 2|V_j|/(\varepsilon k) + 2\log_{1+\varepsilon}(k) - 2/\varepsilon + 7 \right]$$

$$\leq \left( \frac{2}{\varepsilon k} + 1 \right) \sum_{i=1}^{\infty} \mathbb{E}\left[ Y_i(1-\beta\varepsilon)^i n \right] + n + \left( 2\log_{1+\varepsilon}(k) - \frac{2}{\varepsilon} + 7 \right)\ell$$

$$\leq \left( 1 + \left( \frac{2}{\beta\varepsilon} - 2 \right)\left( \frac{2}{\varepsilon k} + 1 \right) \right) n + 4\left( 1 - \frac{1}{\beta\varepsilon} \right)\left( \frac{2(1+\varepsilon)}{\varepsilon}\log(k) - \frac{2}{\varepsilon} + 7 \right)\log(n).$$

Thus, the total queries are $O\left( (1/(\varepsilon k) + 1)n/\varepsilon^3 \right) = O\left( n/\varepsilon^3 \right)$ in expectation. $\square$

# D   Description and Analysis of THRESHOLDSEQ

**Description.** The algorithm takes as input oracle $f$, size constraint $k$, accuracy parameter $\varepsilon > 0$, and probability parameter $\delta$ which influences the failure probability of at most $\delta/n$. The algorithm works in iterations of a sequential outer **for** loop of length at most $O(\log n)$; a set $A$ is initially empty, and elements are added to $A$ in each iteration of the **for** loop. Each iteration has four parts: filtering low value elements from $V$ (Line 5), randomly permuting $V$ (Line 8), computing in parallel the marginal gain of adding blocks of elements of $V$ to $A$ (Line 12), and adding a block slightly larger than the largest block that had average gain at least $(1 - \varepsilon)\tau$ (Line 17).

## D.1   Proof of Theorem 2

### D.1.1   Success Probability

The algorithm THRESHOLDSEQ will successfully terminate once $V$ is empty or $|A| = k$. If it fails to terminate, the outer for loop runs $\ell$ iterations; it also holds that $|V| > 0$ and $|A| < k$ at termination.

**Algorithm 4** A Parallelizable Greedy Algorithm for Fixed Threshold $\tau$

---

1: **procedure** THRESHOLDSEQ($f, \mathcal{N}, k, \delta, \varepsilon, \tau$)
2:     **Input:** evaluation oracle $f : 2^{\mathcal{N}} \to \mathbb{R}^+$, constraint $k$, revision $\delta$, error $\varepsilon$, threshold $\tau$
3:     Initialize $A \leftarrow \emptyset$ , $V \leftarrow \mathcal{N}, \ell = \lceil 4(1 + 2/\varepsilon) \log(n/\delta) \rceil$
4:     **for** $j \leftarrow 1$ to $\ell$ **do**
5:         Update $V \leftarrow \{x \in V : \Delta\left(x \mid A\right) \geq \tau\}$ and filter out the rest
6:         **if** $|V| = 0$ **then**
7:             **return** $A$
8:         $V \leftarrow$ **random-permutation**($V$).
9:         $s \leftarrow \min\{k - |A|, |V|\}$
10:        $\Lambda \leftarrow \{\lfloor (1 + \varepsilon)^u \rfloor : 1 \leq \lfloor (1 + \varepsilon)^u \rfloor \leq s, u \in \mathbb{N}\} \cup \{s\}$
11:        $B \leftarrow \emptyset$
12:        **for** $\lambda_i \in \Lambda$ in parallel **do**
13:           $T_{\lambda_i} \leftarrow \{v_1, v_2, \ldots, v_{\lambda_i}\}$
14:           **if** $\Delta\left(T_{\lambda_i} \mid A\right) / |T_{\lambda_i}| \geq (1 - \varepsilon)\tau$ **then**
15:              $B \leftarrow B \cup \{\lambda_i\}$
16:        $\lambda^* \leftarrow \min\{\lambda_i \in \Lambda : \lambda_i > b, \forall b \in B\}$
17:        $A \leftarrow A \cup T_{\lambda^*}$
18:        **if** $|A| = k$ **then**
19:             **return** $A$
20:     **return** *failure*

---

Fix an iteration $j$ of the outer **for** loop; for this part, we will condition on the behavior of the algorithm prior to iteration $j$.

**Lemma 12.** *At an iteration $j$, let $S_i = \{x \in V : \Delta\left(x \mid T_i \cup A\right) < \tau\}$ after Line 5. It holds that $|S_0| = 0, |S_{|V|}| = |V|$, and $|S_i| \leq |S_{i+1}|$.*

*Proof.* After Line 5, for any $x \in V$, $\Delta\left(x \mid A\right) \geq \tau$, $\Delta\left(x \mid V \cup A\right) = 0$. Since $T_0 = \emptyset$ and $T_{|V|} = V$,

$$
\begin{aligned}
|S_0| &= |\{x \in V : \Delta\left(x \mid A\right) < \tau\}| &&= 0, \\
|S_{|V|}| &= |\{x \in V : \Delta\left(x \mid V \cup A\right) < \tau\}| &&= |V|.
\end{aligned}
$$

Due to submodularity, $\Delta\left(x \mid T_i \cup A\right) > \Delta\left(x \mid T_{i+1} \cup A\right)$. So, $S_i$ is the subset of $S_{i+1}$, which means $|S_i| \leq |S_{i+1}|$. $\qquad\square$

From Lemma 12, there exists a $t$ such that $t = \min\{i \in \mathbb{N} : |S_i| \geq \varepsilon|V|/2\}$. If $\lambda^* \geq t$, we will successfully discard at least $\varepsilon/2$-fraction of $V$ at next iteration. Suppose that the algorithm does not terminate before the outer for loop ends. In the case that $|A| = k$ or $|V| = 0$, we continue the outer for loop with $s = 0$ and select the empty set. Let $i' = \min\{s, t\}$. To fail the algorithm, there should be no more than $m = \lceil \log_{1-\varepsilon/2}(1/n) \rceil$ iterations that $\lambda^* \geq i'$. Otherwise, the algorithm terminates either with $|A| = 0$ or with more than $m$ iterations which successfully filter out at least $\varepsilon/2$-fraction of $V$ resulting in that $|V| = 0$. The following lemma shows that, at each iteration, there is a constant probability to successfully discard at least $\varepsilon/2$-fraction of $V$ or have that $|A| = k$. Then, we will show that the algorithm succeeds with a probability of at least $1 - \delta/n$.

**Lemma 13.** *It holds that $Pr\left(\lambda^* < i'\right) \leq 1/2$, where $i' = \min\{s, t\}$.*

*Proof.* Call an element $v_i \in V$ *bad* iff $\Delta\left(v_i \mid A \cup T_{i-1}\right) < \tau$. Let $\lambda_{i'} = \max\{\lambda \in \Lambda : \lambda < i'\}$. The random permutation of $V$ can be regarded as $|V|$ dependent Bernoulli trials, with success iff the element is bad and failure otherwise. Observe that the probability that an element in $T_{\lambda_{i'}}$ is bad is less than $\varepsilon/2$, condition on the outcomes of the preceding trials. Further, if $\lambda_{i'} \notin B$, there are at least $\varepsilon\lambda_{i'}$ bad elements in $T_{\lambda_{i'}}$, for otherwise, the condition on Line 14 would hold and $\lambda_{i'}$ would be in $B$. Let $(Y_i)$ be a sequence of independent and identically distributed Bernoulli trails, each with success probability $\varepsilon/2$. The probability can be bounded as follows:

$$
\begin{aligned}
Pr\left(\lambda^* < i'\right) &\leq Pr\left(\Delta\left(T_{\lambda_{i'}} \mid A\right)/\lambda_{i'} < (1 - \varepsilon)\tau\right) \\
&\leq Pr\left(\# \text{ of bad elements in } T_{\lambda_{i'}} > \varepsilon\lambda_{i'}\right)
\end{aligned}
$$

$$= Pr\left(\sum_{i=1}^{\lambda_{i'}} \mathbb{1}_{\{v_i \text{ is bad}\}} > \varepsilon\lambda_{i'}\right)$$

$$\leq Pr\left(\sum_{i=1}^{\lambda_{i'}} Y_i > \varepsilon\lambda_{i'}\right) \tag{34}$$

$$\leq 1/2, \tag{35}$$

where Inequality 34 follows from Lemma 7, Inequality 35 follows from Markov's inequality and Law of Total Probability. $\qquad\square$

From the above discussion, to fail the algorithm, there should be no more than $m$ iterations that $\lambda^* \geq i'$. Define a *successful iteration* as an iteration that $\lambda^* \geq i'$. Let $X$ be the number of successes during $\ell$ iterations. From the definition and Lemma 13, $X$ is the sum of $\ell$ dependent Bernoulli random variables, where each variable has a success probability of more than $1/2$. Then, let $Y$ be the sum of $\ell$ independent Bernoulli random variables with success probability $1/2$. Therefore, the failure probability of Algorithm 4 can be bounded as follows:

$$Pr\,(\text{Algorithm 4 fails}) \leq Pr\,(X \leq m)$$
$$\leq Pr\,(Y \leq m) \tag{36}$$
$$\leq Pr\,(Y \leq 2\log(n/\delta)/\varepsilon)$$
$$\leq e^{-\left(\frac{\varepsilon+1}{\varepsilon+2}\right)^2 \cdot \left(1+\frac{2}{\varepsilon}\right)\log\left(\frac{n}{\delta}\right)} \tag{37}$$
$$= e^{-\frac{(\varepsilon+1)^2}{\varepsilon(\varepsilon+2)}\log\left(\frac{n}{\delta}\right)} \leq \delta/n,$$

where Inequalities 36 and 37 follow from Lemmas 6 and 5, respectively.

### D.1.2 Adaptivity and Qeury Complexity

For Algorithm 4, oracle queries incurred by computing the marginal gain on Line 5 and 14. The filtering on Line 5 can be done in parallel. And the inner **for** loop is also in parallel. Therefore, the adaptivity for Algorithm 4 is $O\,(\log(n/\delta)/\varepsilon)$.

With the same notations of $V_j$ and $Y_i$ in Section 2.2, there are no more than $|V_{j-1}| + 1$ and $2\log_{1+\varepsilon}(|V_j|) + 4$ oracle queries on Line 5 and in inner **for** loop, respectively; by Lemma 6 in Appendix A, $\mathbb{E}\,[Y_i] \leq 2$. Then, the expected number of queries is bounded as follows:

$$\mathbb{E}\,[Queries] \leq \sum_{j=1}^{\ell} \mathbb{E}\,\left[|V_{j-1}| + 2\log_{1+\varepsilon}(|V_j|) + 5\right]$$

$$\leq n + \sum_{n(1-\varepsilon/2)^i \geq 1} \mathbb{E}\,[Y_i]\left(n(1-\varepsilon/2)^i + 2\log(n(1-\varepsilon/2)^i)\right) + 5\ell$$

$$\leq n + 2n\sum_{i\geq 1}(1-\varepsilon/2)^i + 4\sum_{n(1-\varepsilon/2)^i \geq 1}\log(n(1-\varepsilon/2)^i) + 5\ell$$

$$\leq n\left(\frac{4}{\varepsilon}+1\right) + 4\log\left(n\left(1-\frac{\varepsilon}{2}\right)\right)\frac{\log_{1-\varepsilon/2}\left(\frac{1}{n}\right)}{2} + 5\lceil 4(1+2/\varepsilon)\log(n/\delta)\rceil.$$

Thus, the total queries are $O\,(n/\varepsilon)$ in expectation.

### D.1.3 The Marginal Gain (Properties 3 and 4 of Theorem 2)

Let $A_j$ be $A$ after iteration $j$, and $T_{j,\lambda_j^*}$ be the subset added to $A$ at iteration $j$. For each $T_{j,\lambda_j^*}$ being added to $A$, there exists a $T_{j,\lambda_j}$, $|T_{j,\lambda_j}| \geq |T_{j,\lambda_j^*}|/(1+\varepsilon)$ and the if condition in Line 14 holds for $T_{j,\lambda_j}$.

$$f(A) = \sum_{j=1}^{\ell} \Delta\left(T_{j,\lambda_j^*} \mid A_{j-1}\right)$$

$$\geq \sum_{j=1}^{\ell} \Delta\left(T_{j,\lambda_j} \mid A_{j-1}\right) \tag{38}$$

$$\geq \sum_{j=1}^{\ell} (1-\varepsilon)\tau |T_{j,\lambda_j}|$$

$$\geq \sum_{j=1}^{\ell} (1-\varepsilon)\tau |T_{j,\lambda_j^*}|/(1+\varepsilon)$$

$$= (1-\varepsilon)\tau/(1+\varepsilon) \cdot |A|,$$

where Inequality 38 follows from monotonicity. Therefore, the average marginal $f(A)/|A| \geq (1-\varepsilon)\tau/(1+\varepsilon)$.

If $|A| < k$ and the algorithm is successful, the algorithm terminates with $|V| = 0$. For any $x \in \mathcal{N}$, $x$ should be filtered out at an iteration $j$ with $A_{j-1}$, which means $\Delta\left(x \mid A_{j-1}\right) < \tau$. Due to submodularity, $\Delta\left(x \mid A\right) \leq \Delta\left(x \mid A_{j-1}\right) < \tau$.

# E    Pseudocode and Omitted Proofs for Section 3

In Algorithm 2, detailed pseudocode of PARALLELGREEDYBOOST is provided. For any set $A$, the notation $f_A(\cdot)$ represents the function $f(A \cup \cdot)$; $f_A$ is submodular if $f$ is submodular.

## E.1    Proof of Theorem 3

### E.1.1    Success Probability

*Proof.* The probability of succeess can be bounded as follows:

$$Pr\left(\text{Algorithm 2 succeed}\right) \geq 1 - Pr\left(\text{THRESHOLDSEQ fails at an iteratin}\right)$$
$$- Pr\left(\alpha\text{-approximation algorithm for SM fails}\right)$$

$$\geq 1 - \sum_{i=1}^{\lceil \log_{1-\varepsilon}(\alpha/3) \rceil} Pr\left(\text{THRESHOLDSEQ fails}\right) - p_\alpha$$

$$\geq 1 - \frac{\delta}{n} \cdot \lceil \log_{1-\varepsilon}(\alpha/3) \rceil - p_\alpha$$

$$\geq 1 - 1/n - p_\alpha.$$

$\square$

### E.1.2    Approximation Ratio

*Proof of Inequality 4.*

$$f(A_{j+1}) - f(A_j) \geq (1-\varepsilon/3)\tau/(1+\varepsilon/3) \cdot |A_{j+1} \backslash A_j| \tag{39}$$

$$\geq \frac{(1-\varepsilon/3)(1-\varepsilon)}{(1+\varepsilon/3)k} |A_{j+1} \backslash A_j| \cdot \sum_{o \in O} \Delta\left(o \mid A_j\right) \tag{40}$$

$$\geq \frac{(1-\varepsilon/3)(1-\varepsilon)}{(1+\varepsilon/3)k} |A_{j+1} \backslash A_j| \cdot (f(O \cup A_j) - f(A_j))$$

$$\geq \frac{(1-\varepsilon/3)(1-\varepsilon)}{(1+\varepsilon/3)k} |A_{j+1} \backslash A_j| \cdot (f(O) - f(A_j)),$$

where Inequalities 39-40 follow from Theorem 2 and $0 \leq |S_j| < k - |A_{j-1}|$.

$\square$

*Proof of Inequality 5.*

$$f(O) - f(A) \leq \prod_{j=0}^{\ell-1} \left(1 - \frac{(1-\varepsilon/3)(1-\varepsilon)}{(1+\varepsilon/3)k} |A_{j+1} \backslash A_j|\right) f(O)$$

$$\leq e^{-\frac{(1-\varepsilon/3)(1-\varepsilon)}{(1+\varepsilon/3)k}\sum_{j=0}^{\ell-1}|A_{j+1}\setminus A_j|}f(O)$$
$$= e^{-\frac{(1-\varepsilon/3)(1-\varepsilon)}{1+\varepsilon/3}}f(O)$$
$$\leq (1/e+\varepsilon)f(O), \tag{41}$$

where Inequality 41 follows from Lemma 14. □

**Lemma 14.** $e^{-\frac{(1-\varepsilon/3)(1-\varepsilon)}{1+\varepsilon/3}} \leq 1/e+\varepsilon$, when $0 \leq \varepsilon \leq 1$

*Proof.* Let $h(x) = 1/e + x - e^{-\frac{(1-x/3)(1-x)}{1+x/3}}$. The lemma holds if $h(x)$ is monotonically increasing on $[0,1]$, since $h(0) = 0$. The remainder of the proof shows that the first derivative of $h(x)$ satisfies that $h'(x) > 0$ on $[0,1]$.

The first and second derivatives of $h(x)$ is as follows:

$$h'(x) = 1 + \frac{x^2+6x-15}{(x+3)^2}e^{-\frac{(3-x)(1-x)}{3+x}},$$
$$h''(x) = \frac{48(x+3)-(x^2+6x-15)^2}{(x+3)^4}e^{-\frac{(3-x)(1-x)}{3+x}}.$$

Let $g(x) = 48(x+3)-(x^2+6x-15)^2$. And, it holds that $g'(x) = 48-4(x+3)(x^2+6x-15) > 0$ when $0 \leq x \leq 1$. Therefore, $g(x)$ is monotonically increasing on $[0,1]$. Since $g(0) = -81 < 0$ and $g(1) = 128 > 0$, $g(x)$ only has one zero $x_0$. And when $0 \leq x \leq x_0$, $g(x) \leq 0$; when $x_0 \leq x \leq 1$, $g(x) \geq 0$. Since $(x+3)^4 > 0$ and $e^{-\frac{(1-x/3)(1-x)}{1+x/3}} > 0$, when $0 \leq x \leq x_0$, it holds that $h''(x) \leq 0$; when $x_0 \leq x \leq 1$, it holds that $h''(x) \geq 0$. Thus, $x_0$ is the minimum point of $h'(x)$. Next, we try to bound the minimum of $h'(x)$.

Since $g(0.361) < 0$ and $g(0.362) > 0$, the zero of $g(x)$ follows that $0.361 \leq x_0 \leq 0.362$. From the analysis above, we know that $|g(x)| \leq \max\{|g(0)|, |g(1)|\} = 128$. Since $(x+3)^4 \geq 81$ and $e^{-\frac{(1-x/3)(1-x)}{1+x/3}} \leq 1$, it holds that $|h''(x)| \leq 128/81$. Therefore,

$$h'(x) \geq h'(x_0) \geq h'(0.361) - \max|h''(x)| \cdot (x_0 - 0.361) > 0.3175 > 0.$$

Thus, $h(x)$ is monotonically increasing on $[0,1]$. It holds that $h(x) \geq h(0) = 0$, when $0 \leq x \leq 1$. □

## F  Lower Adaptivity Modification of LINEARSEQ

In this section, we describe and analyze a variant of LINEARSEQ with lower adaptivity. Pseudocode is given in Alg. 5.

**Theorem 5.** *Let $(f,k)$ be an instance of* SM. *For any constant $0 < \varepsilon < 1/2$, the algorithm* LOWADAPLINEARSEQ *has adaptive complexity $O\left(\log(n/k)/\varepsilon^3\right)$ and outputs $C \subseteq \mathcal{N}$ with $|C| \leq k$ such that the following properties hold: 1) The algorithm succeeds with probability at least $1-k/n$. 2) There are $O\left((1/(\varepsilon k)+1)n/\varepsilon^3\right)$ oracle queries in expectation. 3) If the algorithm succeeds, $\left[5 + \frac{4(2-\varepsilon)}{(1-\varepsilon)(1-2\varepsilon)}\varepsilon\right]f(C) \geq f(O)$, where $O$ is an optimal solution to the instance $(f,k)$.*

*Proof.* The main difference between LOWADAPLINEARSEQ and LINEARSEQ is that the algorithm terminate when $|V| \leq k$ and returns the maximum value between $f(A')$ and $f(V)$. Since the two algorithms have the same procedures of selection and filtering and the value of $\beta$ does not change, the probability of successful filtering of $(\beta\varepsilon)$-fraction of $V$ at any iteration of the outer **for** loop is the same as LINEARSEQ which is at least 1/2.

**Success Probability.** The algorithm LOWADAPLINEARSEQ will succeed if there are at least $m = \lceil \log_{1-\beta\varepsilon}(k/n) \rceil$ successful iterations. By modeling the success of the iterations as a sequence of dependent Bernoulli random variables, and denoting that $X_\ell$ is the number of successes up to and including the $\ell$-th iteration and $Y_\ell$ is the number of successes in $\ell$ independent Bernoulli trails with success probability 1/2,

$$Pr\left(\text{Algorithm 5 succeeds}\right) \geq Pr\left(X_\ell \geq m\right)$$

**Algorithm 5** A lower adaptivity version of LINEARSEQ with $(5 + O(\varepsilon))^{-1}$ approximation ratio in $O\left(\log(n/k)/\varepsilon^3\right)$ adaptive rounds and expected $O\left(n/\varepsilon^3\right)$ queries.

---

1: **procedure** LOWADAPLINEARSEQ($f, \mathcal{N}, k, \varepsilon$)
2:     **Input:** evaluation oracle $f : 2^{\mathcal{N}} \to \mathbb{R}^+$, constraint $k$, error $\epsilon$
3:     $a = \arg\max_{u \in \mathcal{N}} f(\{u\})$
4:     Initialize $A \leftarrow \{a\}$, $V \leftarrow \mathcal{N}$, $\ell = \lceil 4(1 + 1/(\beta\varepsilon))\log(n/k)\rceil$, $\beta = \varepsilon/(16\log(8/(1 - e^{-\varepsilon/2})))$
5:     **for** $j \leftarrow 1$ to $\ell$ **do**
6:         Update $V \leftarrow \{x \in V : \Delta(x \mid A) \geq f(A)/k\}$ and filter out the rest
7:         **if** $|V| \leq k$ **then break**
8:         $V = \{v_1, v_2, \ldots, v_{|V|}\} \leftarrow$ **random-permutation**($V$)
9:         $\Lambda \leftarrow \{\lfloor(1 + \varepsilon)^u\rfloor : 1 \leq \lfloor(1 + \varepsilon)^u\rfloor \leq k, u \in \mathbb{N}\}$
              $\cup\{\lfloor k + u\varepsilon k\rfloor : \lfloor k + u\varepsilon k\rfloor \leq |V|, u \in \mathbb{N}\} \cup \{|V|\}$
10:        $B[\lambda_i] = $ **false**, for $\lambda_i \in \Lambda$
11:        **for** $\lambda_i \in \Lambda$ in parallel **do**
12:            $T_{\lambda_{i-1}} \leftarrow \{v_1, v_2, \ldots, v_{\lambda_{i-1}}\}$ ; $T_{\lambda_i} \leftarrow \{v_1, v_2, \ldots, v_{\lambda_i}\}$ ; $T'_{\lambda_i} \leftarrow T_{\lambda_i} \backslash T_{\lambda_{i-1}}$
13:            **if** $\Delta\left(T'_{\lambda_i} \mid A \cup T_{\lambda_{i-1}}\right)/|T'_{\lambda_i}| \geq (1 - \varepsilon)f(A \cup T_{\lambda_{i-1}})/k$ **then** $B[\lambda_i] \leftarrow$ **true**
14:        $\lambda^* \leftarrow \max\{\lambda_i \in \Lambda : B[\lambda_i] = $ **false** and $((\lambda_i \leq k$ and $B[1]$ to $B[\lambda_{i-1}]$ are all **true**) or $(\lambda_i > k$ and $\exists m \geq 1$ s.t. $|\bigcup_{u=m}^{i-1} T'_{\lambda_u}| \geq k$ and $B[\lambda_m]$ to $B[\lambda_{i-1}]$ are all **true**))$\}$
15:        $A \leftarrow A \cup T_{\lambda^*}$
16:     **if** $|V| > k$ **then return** *failure*
17:     $A' \leftarrow$ last $k$ elements added to $A$
18:     **return** $C \leftarrow \arg\max\{f(A'), f(V)\}$

---

$$\geq Pr\left(Y_\ell \geq m\right) \tag{42}$$
$$\geq 1 - Pr\left(Y_\ell \leq \log(n/k)/(\beta\varepsilon)\right)$$
$$\geq 1 - e^{-\frac{1}{2}\left(\frac{2\beta\varepsilon+1}{2(\beta\varepsilon+1)}\right)^2 \cdot 2\left(1+\frac{1}{\beta\varepsilon}\right)\log(n/k)} \tag{43}$$
$$= 1 - \left(\frac{k}{n}\right)^{\frac{(2\beta\varepsilon+1)^2}{4\beta\varepsilon(\beta\varepsilon+1)}}$$
$$\geq 1 - \frac{k}{n},$$

where Inequalities 42 and 43 follow from Lemma 6 and Lemma 5, respectively.

**Adaptivity and Query Complexity.** Oracle queries are made on Line 6 and 13. There is one adaptive round on Line 6 and also one adaptive round of the inner **for** loop. Therefore, the adaptivity is proportional to the number of iterations of the outer **for** loop, $O(\ell) = O\left(\log(n/k)/\varepsilon^3\right)$.

For the query complexity, let $V_j$ be the value of $V$ after filtering on Line 6, and let $Y_i$ be the number of iterations between the $(i-1)$-th and $i$-th successful iterations of the outer **for** loop. By Lemma 6 in Appendix A, $\mathbb{E}[Y_i] \leq 2$. Observe that $|V_0| = |\mathcal{N}| = n$. Then, the expected number of queries is bounded as follows:

$$\mathbb{E}[Queries] \leq \sum_{j=1}^{\ell} \mathbb{E}\left[|V_{j-1}| + 2|V_j|/(\varepsilon k) + 2\log_{1+\varepsilon}(k) - 2/\varepsilon + 7\right]$$

$$\leq \left(\frac{2}{\varepsilon k} + 1\right)\sum_{i=1}^{\infty} \mathbb{E}\left[Y_i(1 - \beta\varepsilon)^i n\right] + n + \left(2\log_{1+\varepsilon}(k) - \frac{2}{\varepsilon} + 7\right)\ell$$

$$\leq \left(1 + \left(\frac{2}{\beta\varepsilon} - 2\right)\left(\frac{2}{\varepsilon k} + 1\right)\right)n + 4\left(1 - \frac{1}{\beta\varepsilon}\right)\left(\frac{2(1+\varepsilon)}{\varepsilon}\log(k) - \frac{2}{\varepsilon} + 7\right)\log(n/k).$$

The total queries are $O\left((1/(\varepsilon k) + 1)n/\varepsilon^3\right) = O\left(n/\varepsilon^3\right)$ in expectation.

**Approximation Ratio.** Lemma 2 still holds for Algorithm 5, since the selection and filtering procedures do not change. Thus, it also holds that:

$$f(A') \geq \frac{(1 - \varepsilon)(1 - 2\varepsilon)}{2(1 - \varepsilon + \varepsilon^2)}f(A). \tag{44}$$

**Lemma 15.** *Suppose* LOWADAPLINEARSEQ *terminates successfully. Then* $f(O\backslash V) \leq 2f(A)$, *where $O$ is the optimal solution of size $k$.*

*Proof.* For each $o \in O\backslash V$, let $j(o) + 1$ be the iteration where $o$ is filtered out, and $A_{j(o)}$ be the value of $A$ after iteration $j(o)$. It holds that $\Delta\left(o \,|\, A_{j(o)}\right) < f(A_{j(o)})/k$. Then,

$$f(O\backslash V) - f(A) \leq f\left((O\backslash V) \cup A\right) - f(A) \tag{45}$$

$$\leq \sum_{o \in (O\backslash V) \cup A} \Delta\left(o \,|\, A_{j(o)}\right) \tag{46}$$

$$\leq \sum_{o \in (O\backslash V) \cup A} f(A_{j(o)})/k$$

$$\leq f(A), \tag{47}$$

where Inequality 45 follows from monotonicity, and Inequalities 46 and 47 follow from submodularity. $\square$

The approximation ratio can be calculated as follows,

$$f(O) \leq f(O \cap V) + f(O\backslash V) \tag{48}$$

$$\leq f(V) + \frac{4(1 - \varepsilon + \varepsilon^2)}{(1 - \varepsilon)(1 - 2\varepsilon)} f(A') \tag{49}$$

$$\leq \left(1 + \frac{4(1 - \varepsilon + \varepsilon^2)}{(1 - \varepsilon)(1 - 2\varepsilon)}\right) f(C)$$

$$= \left(5 + \frac{4(2 - \varepsilon)}{(1 - \varepsilon)(1 - 2\varepsilon)} \cdot \varepsilon\right) f(C),$$

where Inequality 48 follows from submodularity, and Inequality 49 follows from monotonicity, Inequality 44 and Lemma 15. $\square$

## G   Practical Optimizations to LINEARSEQ

In this section, we describe two practical optimizations to LINEARSEQ that do not compromise its theoretical guarantees. The implementation evaluated in Section 4 uses these optimizations for LINEARSEQ. Full details of the implementation are provided in the source code of the Supplementary Material.

### G.1   Avoidance of Large Candidate Sets $A$

Most of the applications described in Appendix H.2 have runtime that depends at least linearly on $|S|$, the size of the set $S$ to be evaluated. If LINEARSEQ is implemented as specified in Alg. 1, the initial value of $A$ may be arbitrarily low. Therefore, the size of $A$ may grow very large as many low-value elements satisfy the threshold of $f(A)/k$ for acceptance into the set $A$. If $A$ becomes very large, the algorithm may slow down due to the evaluation of many large sets, potentially much larger than $k$.

To avoid this issue, we adopt the following strategy. The universe $\mathcal{N}$ is partitioned into two sets: $\mathcal{N}_1$, consisting of the $5k$ highest value singletons, and $\mathcal{N}_2 = \mathcal{N} \setminus \mathcal{N}_1$. Then the main **for** loop of LINEARSEQ is executed twice; the first time, $V$ is initialized to $\mathcal{N}_1$. The second execution sets $V = \mathcal{N}_2$ and $A$ has the value obtained after the first execution of the **for** loop. After the second execution of the **for** loop, the algorithm concludes as specified in Alg. 1.

The idea behind the first execution of the **for** loop is to obtain an initial set $A$ with a relatively high value; then, when the rest of the elements $\mathcal{N}_2$ are considered, elements with low value with respect to $f(A)$ will be filtered immediately and the size of $A$ will be limited. In fact, one can show that it is sufficient to take $|A| < O(k \log(k))$ using this strategy to ensure that $f(A) \geq$ OPT; we omit this proof, but it follows from an intial value of $f(A)$ of at least the maximum singleton value, the fact that OPT is at most $k$ times the maximum singleton value, and the condition on the growth of the value of $f(A)$ when elements are added.

### G.2 Early Termination

A simple upper bound on OPT may be optained by the sum of the top $k$ singleton values. Hence, the algorithm may check if its ratio is satisfied early, *i.e.*, before the **outer** for loop finishes. If so, the algorithm can terminate early and still ensure its approximation ratio holds.

## H  Experiments

### H.1  Implementation, Environment, and Parameter Settings

The experiments are conducted on a server running Ubuntu 20.04.2 with kernel verison 5.8.0. To efficiently utilize all the available threads, we install the Open-MPI and mpi4py library in the system and implement all the algorithms using Message Passing Interface (MPI) Using MPI we make all the implementations CPU bound as it allows us to minimize communication between the processors by providing explicit control over the parallel communication architecture and the information exchanged between the processors. The experiments are run with the $mpirun$ command and for all the applications, each algorithm is repeated five times and the average of all the repetitions is used for the evaluation of the objective, value, number of query calls and parallel runtime. Similar to Breuer et al. [9], the parallel runtime of the algorithms are obtained by computing the difference in time between two calls to $MPI.Wtime()$, once after all the processors are provided with a copy of the input data and the objective function and the other call at the completion of the algorithm. The objective values used for the evaluation are normalized by the objective value obtained from `ParallelLazyGreedy` [27], an accelerated parallel greedy algorithm that avoids re-evaluating samples that are known to not provide the highest marginal gain. The parameters $\varepsilon$ for LS+PGB is set to $0.1$ and to obtain the $\alpha$ and $\Gamma$ from the preprocessing algorithm LS, the $\varepsilon$ was set at $0.21$. For FAST, the $\varepsilon$ and $\delta$ are set to $0.05$ and $0.025$ respectively, same as the parameter settings in the Breuer et al. [9] evaluation[2].

### H.2  Application Objectives and Datsets

#### H.2.1  Max Cover

The objective of this application for a given graph $G$ and a constraint $k$ is to find a set $S$ of size $k$, such that the number of nodes having atleast one neighbour in the set $S$ is maximized. The application run on synthetic random graphs, each consisting of 100,000 nodes generated using the Erdős–Rényi (ER), Watts-Strogatz (WS) and Barabási–Albert (BA) models. The ER graphs were generated with $p = 0.0001$, while the WS graphs were created with $p = 0.1$ and 10 edges per node. For the BA model, graphs were generated by adding $m = 5$ edges each iteration.

#### H.2.2  Image Summarization on CIFAR-10 data

In this application, given a constraint $k$, the objective is to find a subset of size $k$ from a large collection of images which is representative of the entire collection. The objective of the application used for the experiments is a monotone variant of the image summarization from Fahrbach et al. [19]. For a groundset with $N$ images, it is defined as follows:

$$f(S) = \sum_{i \in N} \max_{j \in S} s_{i,j}$$

where $s_{i,j}$ is the cosine similarity of the $32 \times 32$ pixel values between image $i$ and image $j$. The data set used for the image summarization experiments is the CIFAR-10 test set containing 10,000 $32 \times 32$ color images.

#### H.2.3  Twitter Feed Summarization

The objective of this application for a given constraint $k$ is to select $k$ tweets from an entire twitter feed consisting of large number of tweets, that would represent a brief overview of the entire twitter feed

---

[2]Our code is available at *https://gitlab.com/deytonmoy000/submodular-bestofbothworlds.*

and provide all the important information. The objective and the data set used for this experiments is adopted from twitter stream summarization of Kazemi et al. [22]. For a given twitter feed of $N$ tweets, where each tweet $s \in N$ consists of a set of keywords $W_s$ and $W$ is the set of all keywords in the feed such that $W = \bigcup_{s=1}^{N} W_s$, the objective function for the twitter feed summarization is given by:

$$f(S) = \sum_{w \in W} \sqrt{\sum_{s \in S} \text{score}(w, s)}$$

where $\text{score}(w, s)$ is the number of retweets of the tweet $s$ such that $w \in W_s$, otherwise $\text{score}(w, s) = 0$ if $w \notin W_s$. The experiments for twitter feed summarization uses a data set consisting of 42,104 unique tweets from 30 popular news accounts.

### H.2.4 Influence Maximization on a Social Network.

In influence maximization, the objective is to select a set of social network influencers to promote a topic in order to maximise its aggregate influence. The probability that a random user $i$ will be influenced by the set of influencers in $S$ is given by:

$$f_i(S) = 1 \quad \text{for } i \in S$$
$$f_i(S) = 1 - (1 - p)^{|N_S(i)|} \quad \text{for } i \notin S$$

where $|N_S(i)|$ is the number of neighbors of node $i$ in $S$. We use the Epinions data set with 27,000 users from Rossi and Ahmed [33] for the influence maximization experiments.

### H.2.5 Revenue Maximization on YouTube.

Based on the objective function and data set from Mirzasoleiman et al. [29], the objective for this application objective is to maximise product revenue by choosing set of Youtubers $S$, who will each advertise a different product to their network neighbors. For a given set of users $X$ and $w_{i,j}$ as the influence between user $i$ and $j$, the objective function can be defined by:

$$f(S) = \sum_{i \in X} V \left( \sum_{j \in S} w_{i,j} \right)$$
$$V(y) = y^{\alpha}$$

where $V(S)$, the expected revenue from an user is a function of the sum of influences from neighbors who are in $S$ and $\alpha : 0 < \alpha < 1$ is a rate of diminishing returns parameter for increased cover. We use the Youtube data set consisting of 18,000 users and value of $\alpha$ set to 0.9 for this application.

### H.2.6 Traffic Speeding Sensor Placement.

The objective of this application is to install speeding sensors on a select set of locations in a highway network to maximize the observed traffic by the sensors. This application uses the data from CalTrans PeMS system [1] consisting of 1,885 locations from Los Angeles region.

### H.3 Additional Evaluation Results (continued from Section 4)

Figure 3, 4, 5 and 6 illustrates the performance comparison with FAST across the MaxCover(WS), MaxCover(ER), MaxCover(BA), TweetSumm, InfluenceMax and TrafficMonitor applications. In Fig. 3, we show the mean objective value of FAST and LS+PGB across all the applications and datasets. The objective value is normalized by that of `ParallelLazyGreedy`. Both algorithms perform very similarly with objective value higher than $80\%$ on most instances, however, as demonstrated in Fig. 3(a), 3(b), 3(e) the objective value obtained by FAST is not very stable. Overall as shown in the table 2, LS+PGB either maintains or outperforms the objective obtained by FAST across these applications with the TrafficMonitor and MaxCover (BA) being the instances where it exceeds the average objective value of FAST by $6\%$ and $5\%$ respectively.

Fig.4 demonstrates the mean total queries needed by LS+PGB and FAST for all applications with both FAST and LS+PGB exhibiting a linear scaling behavior with the increasing $k$ values with

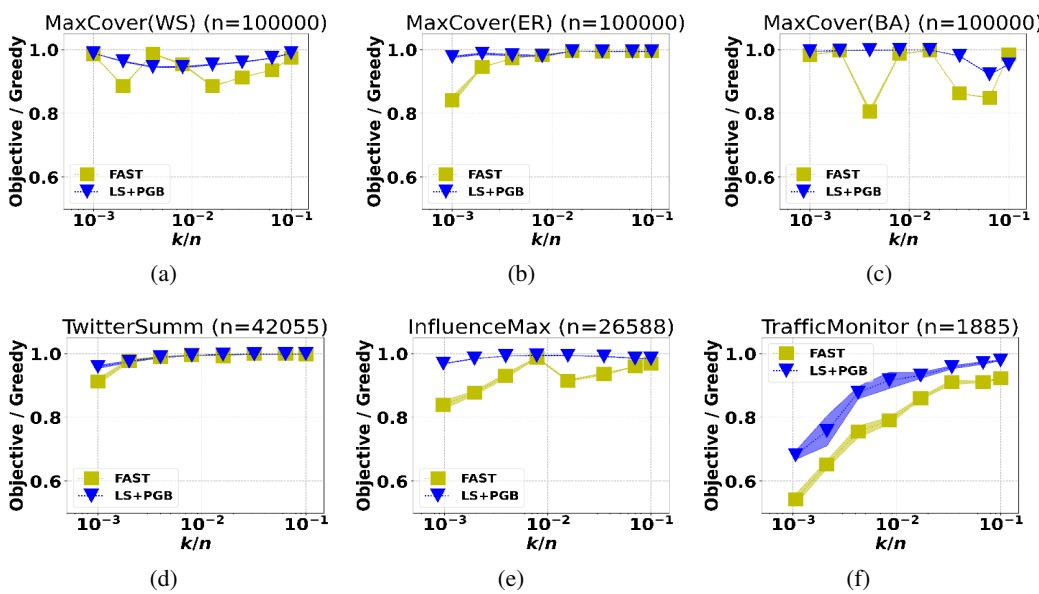

Figure 3: Objective value vs. $k/n$. The objective value is normalized by the standard greedy value. The $(k/n)$-axis is log-scaled.

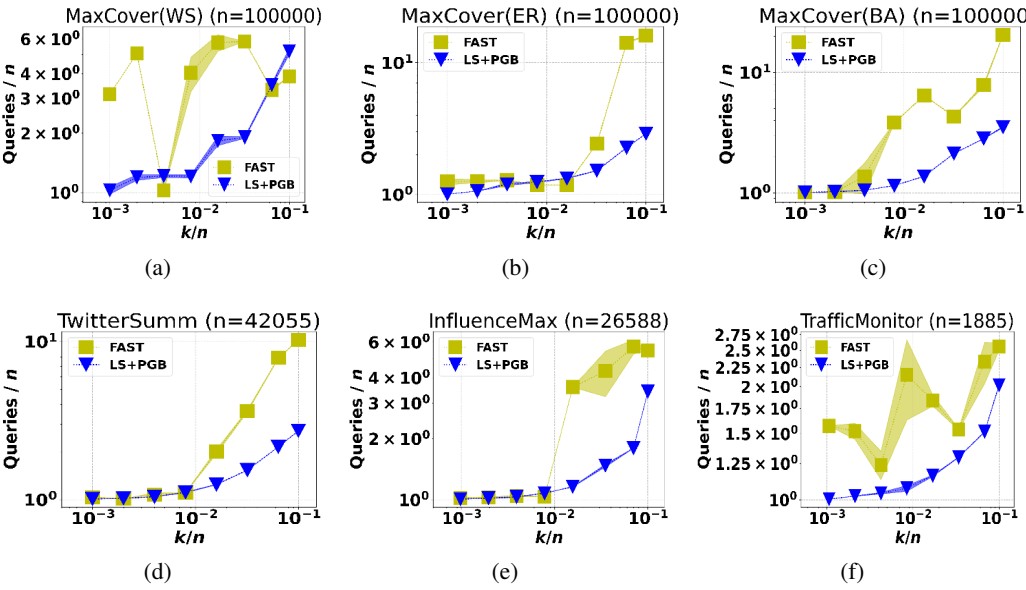

Figure 4: Total Queries/ $n$ vs. $k/n$. Both axes are log-scaled.

the magnitude of rise in total queries with $k$ is less than 5 folds even with 100 folds increase in $k$. Overall as shown in table 2, LS+PGB achieves the objective in less than half the total queries required by FAST for the MaxCover and the TwitterSumm objective. Whereas for TrafficMonitor and InfluenceMax, FAST requires 1.5 and 1.9 times the queries needed by LS+PGB for the same objective. Very similar in nature to the number of query calls, as shown in fig. 6(a) - 6(f), LS+PGB either maintains or outperforms FAST across all the applications.

Fig. 5 and 6 illustrates the adaptivity and the parallel runtime of LS+PGB and FAST across the six datasets. As shown in fig. 6, both algorithms exhibit linear scaling of runtime with $k$. Overall on an average over the six datasets, FAST requires more than 3 times the time needed by LS+PGB to achieve the objective with TwitterSumm being the objective where overall LS+PGB is over 4 times

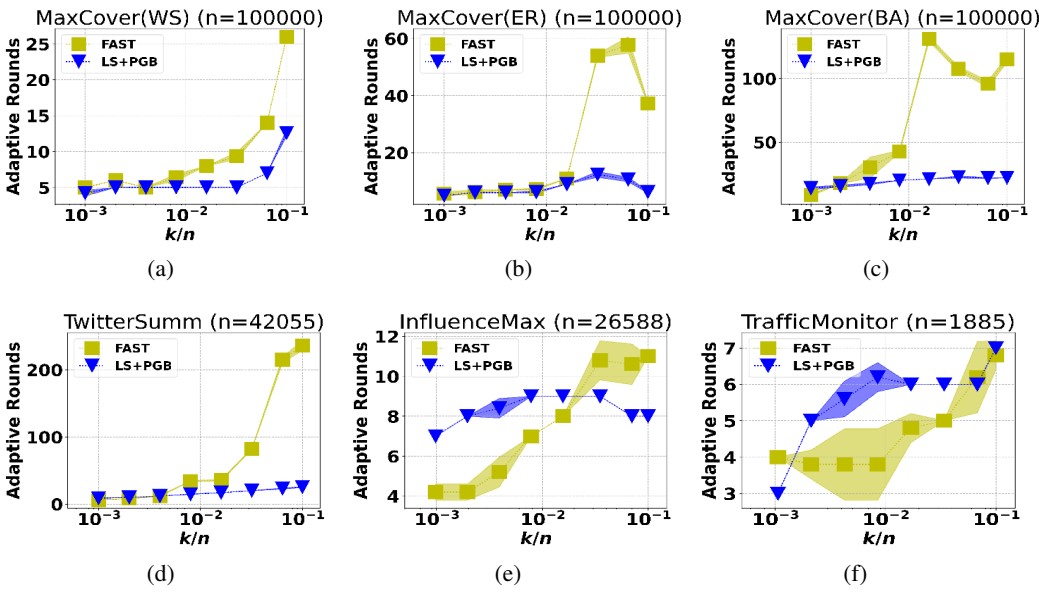

Figure 5: Adaptive Rounds vs. $k/n$. The $(k/n)$-axis is log-scaled.

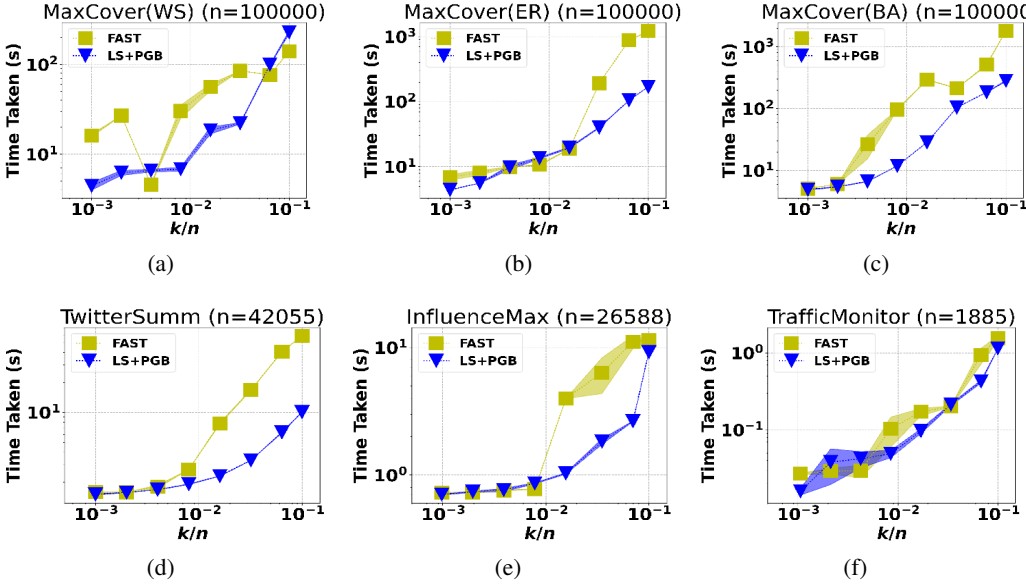

Figure 6: Parallel Runtime vs. $k/n$. Both axes are log-scaled.

quicker than FAST. In terms of adaptivity, as demonstarted in fig. 5, especially for larger values $k$, FAST requires more than double the adaptive rounds needed by LS+PGB for the MaxCover and TwitterSumm application. For InfluenceMax and TrafficMonitor, LS+PGB either maintans or outperforms FAST for larger $k$ values.

## H.4  Time vs. Number of Threads

This experiment set aims to demonstrate the improvement in parallel run time taken by the algorithms to solve the application with increasing number of available threads. We use the SM: maximum cover on random graphs (MaxCover), twitter feed summarization (TweetSumm), image summarization (ImageSumm). See Appendix H.2 for the definition of the objectives. All experiments were conducted with a constant $k$ value of 1,000 and the number of available threads provided to the algorithms

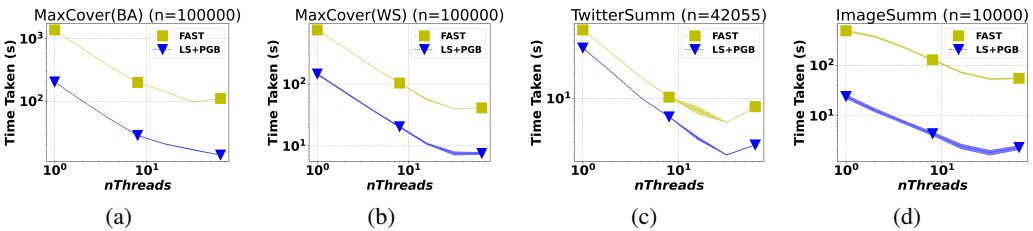

Figure 7: Runtime (s) vs. number of processors. Both axes are log-scaled.

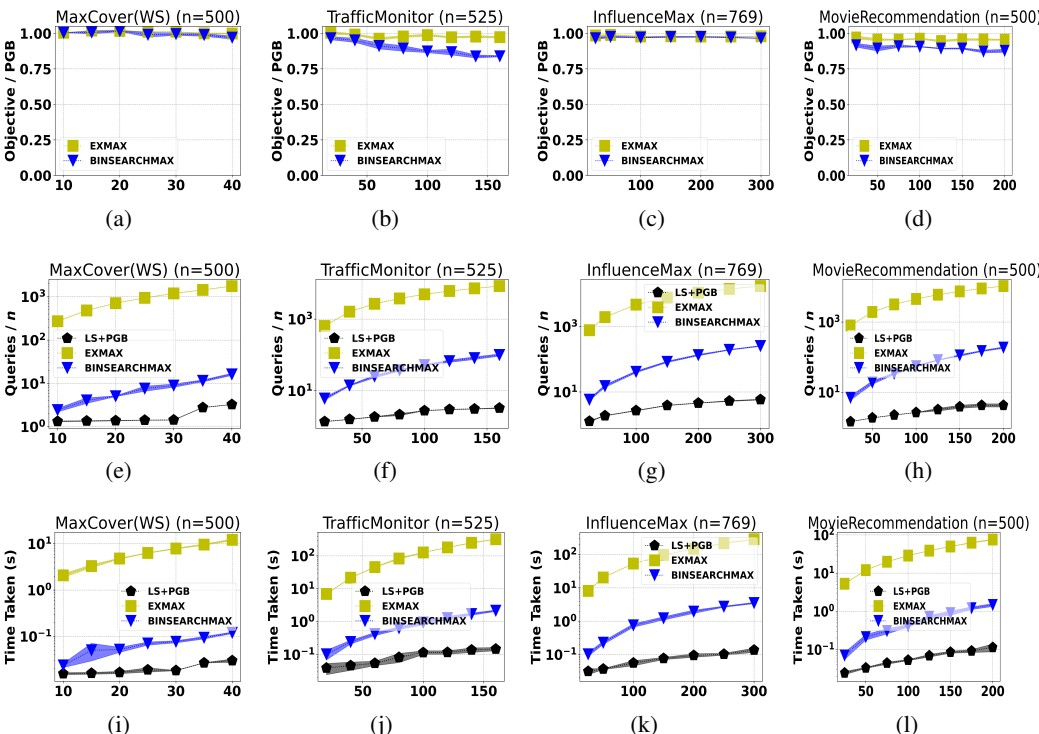

Figure 8: Evaluation of adaptive algorithms from Fahrbach et al. [17] on MaxCover(WS), Traffic-Monitor, InfluenceMax and MovieRecommendation in terms of objective value normalized by the standard greedy value (Figure 8(a) - 8(d)), total number of queries (Figure 8(e) - 8(h)) and the time required by each algorithm (Figure 8(i) - 8(l))

ranged from 1 to 64 threads, doubling the number of threads for each interval in between. The parallel run time of the experiments is measured in seconds. As shown in fig. 7, the scaling behavior of both FAST and LS+PGB are very similar with the number of processors employed, each algorithm exhibiting a linear speedup initially with the number of processors, which plateaus past a certain number of processors. LS+PGB outperforms FAST across all instances with an average speedup of 10 over FAST.

## H.5 Comparison vs adaptive algorithms from Fahrbach et al. [17]

Since EXHAUSTIVEMAXIMIZATION and the BINARYSEARCHMAXIMIZATION algorithm of Fahrbach et al. [17] has better theoretical guarantee than FAST, we ran additional experiments comparing against both the EXHAUSTIVEMAXIMIZATION and the BINARYSEARCHMAXIMIZATION algorithm, as implemented by the authors of FAST. In this highly optimized implementation, a single query per processor is used in each adaptive round instead of the number of queries required theoretically as description of the implementation in Breuer et al. [9]. The evaluation was performed on MaxCover(WS), TrafficMonitor, InfluenceMax and MovieRecommendation in terms of objective

value normalized by the standard greedy value (Figure 8(a) - 8(d)), total number of queries (Figure 8(e) - 8(h)) and the time required by each algorithm. In summary, our algorithm LS+PGB is faster by roughly an order of magnitude and uses roughly an order of magnitude fewer queries over BINARYSEARCHMAXIMIZATION.