# OpenReview forum: "Best of Both Worlds: Practical and Theoretically Optimal Submodular Maximization in Parallel"
_NeurIPS.cc/2021/Conference — NeurIPS 2021 Poster_

### Official Review · Reviewer_1sHC · 2021-07-16

**Rating:** 6
**Confidence:** 5

**Summary:**

In recent years there has been a theoretical breakthrough in low adaptivity O(log(n)) parallel algorithms for submodular maximization, but these algorithms are very slow in practice. Very recently, Breuer et al. proposed an algorithm, FAST, that trades off a small factor in asymptotic adaptive complexity to obtain the practically fastest algorithm. Building on this line of research, the authors propose a new algorithm with slightly better practical performance absent the tradeoff in asymptotic adaptive complexity. The authors demonstrate the practical performance advantage by benchmarking their approach against the FAST algorithm on numerous experiments.


**Ethics Review Area:**

["I don’t know"]

**Limitations And Societal Impact:**

The limitations are noted; the societal impact is not applicable.

**Main Review:**

Overall, I agree with the authors that it is desirable to design a practically fast algorithm with O(log(n)) adaptivity. Some of the ideas used to obtain this, while based significantly on recent work, are also nontrivial. For example, the LinearSeq subroutine and also the ThresholdSeq subroutine to identify high value elements are of general interest and might be used to accelerate other algorithms. In general, the ideas here refine and expand upon the sequencing techniques introduced in Balkanski et al., 2020 and Breuer et al., 2020. The paper and proofs are clearly written.

However, the magnitude of the contribution of the paper hinges on the claim that the proposed algorithm is empirically faster than the FAST algorithm. The reported runtimes and objective values suggest that the algorithm is slightly faster for most (but not all) experiments and parameter combinations. This may be thought of as incremental progress, and to contextualize the magnitude of the obtained speedups, I would note that it is possible to achieve similar or greater speedups for the FAST algorithm by considering various factors such as e.g. better parallel load balancing, etc. (though these might also apply to the authors' algorithm). I will nonetheless acknowledge that if the authors' claims are correct, then their algorithm is currently the fastest for generic submodular maximization (even if by a small margin), which is still useful/interesting. Also, I note that the reported query counts do appear to significantly improve on FAST's queries for most experiments (Appendix Fig. H.3). This means that the performance advantage may actually be better on different (larger?) hardware.

Because the magnitude of the contribution hinges on the experimental results, implementations and experimental setup are critical.

In this respect, I appreciate that the majority of the applications replicate Breuer et al (the FAST algorithm), and also that the authors have also added the Image Summary and Information Monitoring objectives to add some diversity. This gives a nice balance of 6 established benchmarks (from the FAST paper) and two additional ones for diversity. The plots are clear; the values of k are chosen and plotted in a theoretically reasonable way (as values of k/n), and the authors show averages over multiple runs + error bars.

I also dug around the codebase (which is nice to see in the submission). It appears to extend the FAST codebase, and it uses MPI functions, helper functions, etc. from the former. I also develop in MPI, and I appreciate that this is nontrivial. It is also advantageous here because it gives confidence that the authors' comparison to FAST uses a fair/coonosistent comparison of codes, subroutines, timings, etc. It is also nice to see papers adopting and expands a fast parallel codebase & benchmarks that can be used by other researchers in the future.

I would note that because the hardware appears to be a server machine instead of AWS/GoogleCloud/Azure, the reported timings are not strictly replicable (absent buying an identical server machine). As a heuristic check, I compared the benchmark runtimes the authors obtain using FAST against those reported in the original FAST paper, and they do appear to be reasonably consistent. (Note: Please consider adding a \ref to Appendix H in the text of the paper where you discuss threads/processors.)

One oddity in the experiments is that the Parameters paragraph (line 328) and also Appendix H state "these are the same parameter settings for FAST as in the Breuer et al. ", but Breuer et al. (p.6) reports using probability 0.95, not 0.975 as reported here. 0.975 is a fine value to use for benchmarking, but consider revising these paragraphs in the paper and the Appendix (as they also use the probability parameter to compute the values of k under which the Breuer paper's guarantees are heuristic only.)

I will consider updating my review based on the authors' response.

**UPDATE after reading the authors' response**
Thank you for the response. I had a closer look at the code and experiments. There are two things that I think would improve the paper and code. First, I would just re-emphasize that you ought to consider benchmarking on AWS or a cloud computing platform such that your experimental results can be shown to not be dependent on your specific lab hardware. Second, I note the following line of your code inside your parallel_adaptiveAdd() method (which shoulders a lot of the computation for your algorithm):

line 148:             gain= objective.value( tmpS ) - valTmpS;
 which can be found inside your parallel_adaptiveAdd() method.

This is technically fine, but if the goal is to do a fair comparison with the FAST code, consider instead:

gain= objective.marginalval( new_tmpS, tmpS);  # where new_tmpS = list( set(tmpS) | set( Ti) )

The reason is that the FAST codes allow you to specify a marginalval() method, which might use memoization or other speedups. Your code in contrast is memoizing the value of the original  tmpS (once per processor), but computing the full function value of the new tmpS and taking the difference of the two.

The point is that depending on the specific objective function, and also on how objective.marginalval() is written (e.g. whether it uses memoization or speedups on a particular objective), either the FAST code or your code will have a potentially significant advantage. Fixing this will allow the two to be on the same footing.

Again, I do not believe this should lower your review score as it could either help or hurt your algorithm's performance vis-a-vis FAST depending on the specific objective function---I merely suggest it as a means to improve the fairness of the comparison.




**Time Spent Reviewing:**

4.5

---

> ### Author Response · Authors · 2021-08-09
> **Author Response**
>
> Thank you for the valuable feedback. We will follow the suggestions made concerning the presentation.
> - R4: *One oddity in the experiments is that the Parameters paragraph (line 328) and also Appendix H state "these are the same parameter settings for FAST as in the Breuer et al. ", but Breuer et al. (p.6) reports using probability 0.95, not 0.975 as reported here. 0.975 is a fine value to use for benchmarking, but consider revising these paragraphs in the paper and the Appendix.*
>
>   **Answer:** Thank you for catching this. We had originally intended to use exactly the same parameter settings as in Breuer et al., but the code provided by Breuer et al. actually sets $\delta = \epsilon$. We decided not to change the FAST code even in the slightest, which meant that we could not run with the same parameter values $(0.025, 0.05)$ as in Breuer et al. Instead, we set $\epsilon = \delta = 0.025$, which matches the value of $\epsilon$ of Breuer et al., but not $\delta$. We will update the statements in the text to clarify that we did not use the same $\delta$ value as Breuer et al.
>
> - R4: *I would note that it is possible to achieve similar or greater speedups for the FAST algorithm by considering various factors such as e.g. better parallel load balancing, etc. (though these might also apply to the authors' algorithm)*
>
>   **Answer:** We agree it may be possible to improve practical runtime of the algorithms further by better load balancing, but we would expect to see a similar benefit to both our algorithm and FAST for the following reason. Both algorithms currently use the same strategy (and indeed the same code when possible) to divide queries between threads -- the set $X$ of queries in the current adaptive round is divided into equal parts among the threads. By reassigning some queries to threads that process their workload more quickly than others, it may be possible to obtain further performance gains for both algorithms.
>
>   Moreover, our algorithm would likely retain its performance advantage after optimizing the algorithms in this way, as evidenced by the improvement in the total number of queries required by our algorithm (Figure H.3).

---

### Official Review · Reviewer_ubfR · 2021-07-16

**Rating:** 6
**Confidence:** 4

**Summary:**

This work introduces the LinearSeq subroutine, which achieves a 1/4
approximation ratio for monotone submodular maximization subject to a
cardinality constraint w.h.p. while using $O(n)$ queries in expectation and
$O(\log n)$ adaptive rounds. The authors then show how to use LinearSeq
together with a thresholding subroutine to achieve an algorithm with a
$(1-1/e-\varepsilon)$ approximation for monotone submodular maximization in
$O(1/\varepsilon^2 \log(n / \varepsilon))$ adaptive rounds and an expected $O(n
/ \varepsilon^2)$ queries. This is an improvement over the previous works of
[Balkanski-Rubinstein-Singer, SODA 2019], [Ene-Nguyen, SODA 2019],
[Fahrbach-Mirrokni-Zadimoghaddam, SODA 2019] in terms of $\varepsilon$ factors
(for the query complexity) and practicability.

The authors compare their work extensively to the recent FAST algorithm of
[Breuer-Balkanski-Singer, ICML 2019], which sacrifices some theoretical
guarantees in favor of being extremely practical. Compared to all previous
works, the main result here improves the query complexity without sacrificing
the approximation quality or (much of) the adaptivity. The experiments in this
paper compare against FAST and demonstrate that the algorithm in this paper is
substantially faster than FAST without compromising on the solution quality
(see Figure 3 in the full version of the paper).

**Main Review:**

**Originality.**
This paper builds on recent line of work on low-adaptivity monotone submodular
maximization. The overarching approach is reasonably well understood in the
submodular literature, but the new subroutine (LinearSeq) that the authors
introduce is a powerful, standalone technique for preprocessing the candidate
space for a good guess of OPT, and therefore can be used to reduce the query
complexity of all existing low-adaptivity algorithms (i.e., this idea allows us
to find an interval $[L, U]$ such that $U/L = O(1)$ and $OPT \in [L, U]$ using
a nearly optimal number of queries and adaptive rounds).

**Quality.**
The LinearSeq subroutine is a low-adaptive adaptation of the novel streaming
algorithm of [Kuhnle, AISTATS 2021]; hence, this paper does a great job of
bridging the two literatures (streaming and low-adaptivity) in a meaningful
way. Overall, the paper is good, but it's not clear that the previous
state-of-the-art algorithm for this problem is the FAST algorithm
[Breuer-Balkanski-Singer, ICML 2020], even though it's the most recent in this
line of work. The authors compare only against this, but it would be valuable
to see how the algorithms from SODA 19 compare as well, as they have better
theoretical guarantees than FAST.

The theory is interesting and seems to follow the high-level approach of
[Fahrbach-Mirrokni-Zadimoghaddam, SODA 2019], which is a thresholding
batch-greedy algorithm. The experiments could be improved in my opinion, since
they only compare against FAST. In the first nine pages, only the running times
are considered, which can be somewhat implementation-specific. For the revision,
it seems that comparing the solution quality (i.e., Figure 3 in the appendix)
would be more meaningful, since most of the running time information is
captured in Table 2. Further, the drop in running time for FAST in the
MaxCover(WS) experiment seems inconsistent with the other data points and
should probably be explained (especially since this is averaged over five
trials). In general, the running time of this algorithm is 10x faster than FAST
with comparable solution quality and ~10x fewer oracle queries.

**Clarity.**
The paper is written pretty well for a familiar audience. One comment here is
that I would advertise all results of this paper clearly in the abstract, not
just that LinearSeq achieves a 1/4 approximation with nearly optimal adaptivity
and query complexity. At quick glance, it's not clear that the paper contains a
$(1-1/e-\varepsilon)$-approximation algorithm, which then makes the optimality
component of the title somewhat confusing.

**Significance.**
Overall, this is a good result in an active area of research. The remaining
problems in low-adaptivity monotone submodular maximization are (1) shaving off
$\varepsilon$ factors and (2) making the algorithms more practical. This paper
tackles both of these problems.

**Typos / Suggestions.**
- Suggestion: Consider removing "Best of Both Worlds" from the title since it
  somewhat implies that no previous works achieve this balance.
- [33] Suggestion: Consider using the author names + the citation in the
  Reference column of Figure 1. It seems like this would be easier to read than
  the algorithm acronym and citation number.
- [74] Suggestion: Would be good to include the line of low-adaptivity
  submodular maximization algorithms for non-monotone functions, instead of just
  citation [22]:
  * Balkanski, Eric, Adam Breuer, and Yaron Singer. "Non-monotone submodular
    maximization in exponentially fewer iterations." Advances in Neural
    Information Processing Systems 31 (NeurIPS 2018).
  * Fahrbach, Matthew, Vahab Mirrokni, and Morteza Zadimoghaddam. "Non-monotone
    submodular maximization with nearly optimal adaptivity and query complexity."
    International Conference on Machine Learning. PMLR, 2019.
  * Ene, Alina, and Huy Nguyen. "Parallel algorithm for non-monotone
    DR-submodular maximization." International Conference on Machine Learning.
    PMLR, 2020.
- [93] Suggestion: The paper seems to have a somewhat of an aggressive in
  the first few pages (in my opinion). The superior results are clear from the
  main theorem alone, so I'd let them speak for themself. Examples:
  * [93] "arguably, this is the right way to conduct adaptive sequencing"
  * [77] "highly impractical" -- maybe, but it's probably better to say "very expensive"
- [158] Typo: Line 2 of Algorithm 1 uses $\epsilon$ instead of $\varepsilon$
- [280] The guarantees of ThresholdSeq in Theorem 2 has the same format of
  Lemma 3.2 in [Fahrbach-Mirrokni-Zadimoghaddam, SODA 2019] for their
  Threshold-Sampling algorithm. If the ThresholdSeq algorithm was inspired by
  that subroutine, it makes sense to at least point to that reference and their
  algorithm.
- [302] Is the $\alpha$ in "$\Gamma$, $\alpha$, where the..." a typo?

**Time Spent Reviewing:**

3

---

> ### Author Response · Authors · 2021-08-09
> **Author Response**
>
> Thank you for the valuable feedback. We respond to several of the points raised.
> - R3: *The authors compare only against [FAST], but it would be valuable to see how the algorithms from SODA 19 [FMZ19] compare as well, as they have better theoretical guarantees than FAST.*
>
>   **Answer:** To address this concern, we ran additional experiments comparing against both the ExhaustiveMaximization and the BinarySearchMaximization algorithm of [FMZ19], as implemented by the authors of FAST. In this highly optimized implementation, a single query per processor is used in each adaptive round instead of the number of queries required theoretically (see the description of the implementation in Breuer et al. 2020).
>   Please see the results at this anonymized [link](https://ufile.io/ktorrb4a) (Please select "Free Download"). In summary, our algorithm LS+PGB is faster by roughly an order of magnitude and uses roughly an order of magnitude fewer queries over BinarySearchMaximization. These new experiments will be added to the next version of our manuscript.
> - R3: *In the first nine pages, only the running times are considered, which can be somewhat implementation-specific. For the revision, it seems that comparing the solution quality (i.e., Figure 3 in the appendix) would be more meaningful, since most of the running time information is captured in Table 2.*
>
>   **Answer:** We agree that Table 2 well captures the runtime information, and that results concerning solution quality should be moved from the supplementary appendices to the main text in the next version. Thank you for the suggestion.
> - R3: *If the ThresholdSeq algorithm was inspired by that subroutine [FMZ19], it makes sense to at least point to that reference and their algorithm.*
>
>   **Answer:** [FMZ19] was indeed the first to present a subroutine for the subproblem of adding all elements of gain a constant $\tau$ in lowly adaptive manner. We have limited discussion of prior work on this subproblem on line 71, but we will include additional discussion in the ThresholdSeq section in our next revision.
>
> - R3: *The drop in running time for FAST in the MaxCover(WS) experiment seems inconsistent with the other data points and should probably be explained (especially since this is averaged over five trials).*
>
>   **Answer:** FAST employs practical optimization to their code that enable early termination if certain conditions are met. For example, the FAST algorithm terminates if it ever finds a solution better than $(1 - 1/e) \Lambda$, where $\Lambda$ is the sum of the top $k$ singleton values.
>   The fluctuation in the WS runtime is a very good example of the inconsistency this early stopping criterion causes -- for the revision, we will include a detailed explanation of the MaxCover (WS) fluctuations.
>
> - We will carefully follow all suggestions concerning tone and references in the revision. Thank you for the suggestions.

---

### Official Review · Reviewer_EF6P · 2021-07-16

**Rating:** 8
**Confidence:** 4

**Summary:**

This paper proposes several new algorithms for maximizing submodular functions which when combined achieve state of the art combination of approximation ratio, adaptive complexity, and query complexity. Empirically, the combined LS+PGB algorithm is also equal to or better than the recent FAST algorithm.


**Limitations And Societal Impact:**

Little to no discussion of limitations or societal impact. The authors could discuss environmental impact of large-scale datasets and/or subset selection problems reinforcing existing biases in data

**Main Review:**

This paper proposes a new algorithm for maximizing submodular functions that achieves state of the art combination of approximation ratio, adaptive complexity, and query complexity. Empirically, the algorithm is also equal to or better than FAST.

Impact: This is nice paper with impressive results that improve over previous state of the art. The improved ThresholdSeq algorithm in particular may improve many existing algorithms that rely on this subproblem.

Quality/Originality: Builds on and expands previous work in a novel way. The experiments would be improved if additional large scale datasets were used such as ImageNet, MovieLens, or Uber Pickups

Clarity/Organization:
- Very good overview of previous state of the art, with guarantees as well as high level explanation of various algorithms
- The paper ends abruptly with many results deferred to the Supplementary material. I suggest moving the complete proof of ThresholdSeq to the main paper as a warmup, then explain how the analysis must change for LinearSeq and defer the full proof of LinearSeq to the Supplementary Material.
- Please mention that "adaptivity", "adaptive complexity", and "adaptivity complexity" are used interchangeably

Questions:
- How does the proposed method compare to FAST for larger values of k?

---
EDIT: The authors answered my main question in their response. Therefore, I am keeping my review the same.

**Time Spent Reviewing:**

4

---

> ### Author Response · Authors · 2021-08-09
> **Author Response**
>
> Thank you for the valuable feedback. We agree and will follow the suggestions concerning the organization of the paper in the preparation of the next version.
> - R2: *How does the proposed method compare to FAST for larger values of k?*
>
>   **Answer:** In the paper, the maximum $k$ value considered was $n/10$ on each application.  To answer this question, we ran each application with $k=n/2$; please see the results in Table 1 on the second page of the file found at this anonymized [link](https://ufile.io/ktorrb4a) (Please select "Free Download"). The results in comparison with FAST are qualitatively similar to the results reported in the paper.

---

### Official Review · Reviewer_gBDP · 2021-07-20

**Rating:** 7
**Confidence:** 4

**Summary:**

Submodular maximization is an important problem with many relevant applications in machine learning. With the huge amount of existing data and increase in the available computational power (increase in the number of machines), it is important to design scalable and parallelizable algorithms with provable guarantees from both practical and theoretical points of view. Many recent works have studied the submodular maximization in the context of adaptivity (“The degree of parallelizability can be measured by the adaptive complexity of an algorithm, which is the minimum number of rounds into which the queries to f may be organized”). In this paper, the authors propose an algorithm that achieves the state-of-the-art in terms of adaptive complexity, query complexity, and approximation ratio.

**Limitations And Societal Impact:**

I do not see any immediate negative societal impact from this work.

**Main Review:**

This is a nicely written paper. It studies an important paper. It adds another important piece to the collection of recent developments in the scalable submodular maximization domain. The authors have performed an extensive set of experiments. The code and the data for the experiments are shared. As far as I checked, the proof looks correct. As a result, I recommend this paper for acceptance.

I have a few suggestions that could improve the presentation of the paper:
* It would be useful to add FAST [8] to Table 1 as it is used as the baseline in the experiments.
* In the experiments, it is useful to report the adaptivity of the algorithms.
* Adding more details to the experiments in the main text could improve comparison. You can reduce the number of datasets and instead report the objective, adaptivity, and number of queries too.
* You can add more context to the explanation of the main algorithm.

**Time Spent Reviewing:**

5

---

> ### Author Response · Authors · 2021-08-09
> **Author Response**
>
> Thank you for the valuable feedback. We agree that all four of the suggestions would improve the quality of the presentation. We will carefully follow these suggestions in the next version of the paper.

---

### Decision · Program_Chairs · 2021-09-27

**Decision:**

Accept (Poster)

**Comment:**

The reviewers liked both the theoretical guarantees and the empirical performance achieved by the algorithm proposed in this paper, which improves over state-of-the-art algorithms for submodular maximization. For the final version, please include the additional comparison to [FMZ19] that was discussed in the response.